# A chromatin-remodeling-independent role for ATRX in protecting centromeric cohesion

Lei Zhao [ID][1,2], Xueying Yuan[1,3], Qinfu Chen [ID][1], Haiyan Yan [ID][4✉] & Fangwei Wang [ID][1,3,5✉]

## Abstract

**Sister-chromatid cohesion mediated by the cohesin complex is critical for accurate chromosome segregation during mitosis. A key aspect of this process is the protection of cohesin at mitotic centromeres to resist spindle pulling-forces until anaphase onset. However, the mechanisms that prevent cohesin removal by its release-factor Wapl at centromeres remain incompletely understood. In this study, we identify ATRX, a chromatin remodeler of the SWI/SNF family, as a new binding protein of the cohesin complex. ATRX directly interacts with the cohesin accessory subunit Pds5B, antagonizing Wapl binding and thereby preventing premature release of centromeric cohesin. A mutation in ATRX that disrupts its interaction with Pds5B weakens centromeric cohesion and increases chromosome missegregation. Notably, centromere tethering of a Pds5B-binding fragment of ATRX, which lacks the ATPase domain, rescues cohesion defects in ATRX-depleted cells. Furthermore, Wapl depletion bypasses the requirement for ATRX, underscoring their antagonistic relationship. Together, these findings reveal a chromatin-remodeling-independent role for ATRX in maintaining centromeric cohesion by competitively inhibiting Wapl, providing new insights into the mechanisms that safeguard genomic stability.**

**Keywords** Mitosis; Centromere; Sister Chromatid Cohesion; Cohesin; ATRX
**Subject Categories** Cell Cycle; Chromatin, Transcription & Genomics; DNA Replication, Recombination & Repair

## Introduction

During the eukaryotic cell cycle, chromosomes are duplicated in S phase and subsequently segregated during mitotic anaphase, ensuring that each daughter cell receives an identical copy of the genome. The fidelity of chromosome transmission critically depends on the maintenance of sister chromatid cohesion, which is established during DNA replication and preserved until anaphase

onset in late mitosis (Peters and Nishiyama, 2012). Premature loss of sister chromatid cohesion can lead to chromosome missegregation and aneuploidy, conditions strongly associated with cancer (Gordon et al, 2012).

Sister chromatid cohesion is mediated by the ring-shaped cohesin complex, which topologically entraps sister chromatids to keep them together (Guacci et al, 1997; Michaelis et al, 1997). The cohesin complex is composed of core subunits SMC1A, SMC3, RAD21 (also called Scc1 or Mcd1), and either SA1 or SA2, along with the accessory subunit Pds5 (Fig. EV1A) (Nasmyth and Haering, 2009). Pds5 plays both positive and negative roles in regulating cohesion (Hartman et al, 2000; Panizza et al, 2000), with the regulatory subunits Wapl and Sororin competing for interaction with Pds5 to modulate the dynamic association between cohesin and chromatin (Nishiyama et al, 2010; Ouyang et al, 2016; Peters and Nishiyama, 2012).

In vertebrate cells, cohesin is loaded onto chromatin during mitotic telophase and early G1 phase, a process that may involve the opening of the interface between the hinge domains of SMC1A and SMC3 (Buheitel and Stemmann, 2013; Gruber et al, 2006; Murayama and Uhlmann, 2015), although the exact mechanism requires further investigation. During G1, the association of cohesin with chromatin is highly dynamic, as Wapl, in complex with Pds5, can release cohesin from chromatin (Chan et al, 2012; Gandhi et al, 2006; Haarhuis et al, 2013; Kueng et al, 2006; Shintomi and Hirano, 2009; Sutani et al, 2009; Tedeschi et al, 2013). Wapl facilitates the opening of the cohesin ring by disrupting the SMC3-RAD21 interface, leading to cohesin release (Buheitel and Stemmann, 2013; Eichinger et al, 2013). In S phase, acetylation of SMC3 promotes the recruitment of Sororin to Pds5 (Lafont et al, 2010; Nishiyama et al, 2010; Rankin et al, 2005; Schmitz et al, 2007), which competitively displaces Wapl from Pds5, thereby inhibiting Wapl's cohesin-releasing activity and stabilizing a fraction of cohesin on chromatin to allow the establishment of cohesion between newly replicated sister chromatids (Nishiyama et al, 2010; Ouyang et al, 2016).

As cells enter mitosis, cohesin is progressively released from condensing chromosomes in a stepwise manner (Waizenegger et al, 2000). Starting in prophase, phosphorylation of Sororin by the mitotic kinases Aurora B and Cdk1 (Dreier et al, 2011; Liu et al, 2013b; Nishiyama et al, 2013), along with phosphorylation of Pds5 by Nek2a and cyclin A2-Cdk1/2 (Hellmuth and Stemmann, 2024),

[1]Department of Gynecologic Oncology, Women's Hospital, School of Medicine and MOE Laboratory of Biosystems Homeostasis & Protection, Life Sciences Institute, Zhejiang University, Hangzhou, China. [2]Zhejiang Key Laboratory of Molecular Cancer Biology, Life Sciences Institute, Zhejiang University, Hangzhou, China. [3]Zhejiang Key Laboratory of Geriatrics and Geriatrics Institute of Zhejiang Province, Affiliated Zhejiang Hospital, Zhejiang University School of Medicine, Hangzhou, China. [4]Zhejiang Key Laboratory of New Targets and Drugs for Nerve Injury Repair, School of Medicine, Hangzhou City University, Hangzhou, China. [5]State Key Laboratory of Transvascular Implantation Devices, Hangzhou, China. ✉E-mail: yanhy@hzcu.edu.cn; fwwang@zju.edu.cn

displaces Sororin from Pds5 and allows Wapl to bind Pds5, resulting in the release of most cohesin from chromosome arms. Additionally, phosphorylation of SA2 by Plk1 facilitates Wapl-mediated removal of cohesin from chromosome arms (Hauf et al, 2005; Nishiyama et al, 2013). However, a small fraction of cohesin remains protected from the Wapl-Pds5 complex at the centromere (Haarhuis et al, 2014). This centromeric cohesin is destroyed only at anaphase onset by Separase-mediated cleavage of RAD21, allowing sister chromatids to move toward opposite spindle poles (Uhlmann et al, 1999).

Retention of centromeric cohesin prior to anaphase not only counteracts spindle pulling forces to prevent premature chromatid separation, but also enables tension sensing to promote chromosome bi-orientation and accurate segregation (Haarhuis et al, 2014; Mirkovic and Oliveira, 2017; Ruiz et al, 2024). Despite this, the mechanisms that antagonize the Wapl-Pds5 interaction at mitotic centromeres remain incompletely understood. During prometaphase, the shugoshin-1 (Sgo1) protein is enriched at the inner centromere region (Kitajima et al, 2006; McGuinness et al, 2005; Tang et al, 2006), where it binds the cohesin complex at the composite interface between RAD21 and SA2 (Garcia-Nieto et al, 2023; Hara et al, 2014; Liu et al, 2013b; Yan et al, 2024; Yuan et al, 2024). The Sgo1-bound protein phosphatase 2A (PP2A) counteracts Aurora B/Cdk1-mediated phosphorylation of Sororin (Liu et al, 2013b; Nishiyama et al, 2013), preserving the Sororin-Pds5 interaction at centromeres. However, upon bipolar attachment of sister kinetochores to spindle microtubules, the Sgo1-PP2A complex redistributes from the inner centromere to kinetochore-proximal regions well before anaphase onset (Lee et al, 2008; Liu et al, 2013a; Liu et al, 2015), likely deprotecting the Sororin-Pds5 interaction at the inner centromere. This suggests that either Wapl becomes globally inactive during metaphase, or that additional factors are involved in antagonizing the Wapl-Pds5 interaction specifically at mitotic centromeres.

In this study, we identify ATRX (alpha-thalassemia mental retardation X-linked), an ATP-dependent chromatin remodeler of the SWI/SNF family (Aguilera and López-Contreras, 2023), as a novel binding partner of cohesin. We show that ATRX associates with the cohesin complex through direct binding to Pds5B, a paralog of Pds5 that is crucial for centromere cohesion. The ATRX-Pds5B interaction antagonizes Wapl binding to Pds5B at centromeres, thereby protecting centromeric cohesion and ensuring accurate chromosome segregation during mitosis.

## Results

### A proteomic screen identifies ATRX as a Pds5B-associated factor

Vertebrates express two Pds5 proteins, Pds5A and Pds5B (Losada et al, 2005; Sumara et al, 2000). While both are involved in regulating telomere and arm cohesion, Pds5B is specifically required for centromeric cohesion (Carretero et al, 2013). To investigate how the Wapl-Pds5B interaction is antagonized at mitotic centromeres, we performed an unbiased proteomic screen to identify proteins that bind Pds5B. Myc-tagged Pds5B (Myc-Pds5B) was immunoprecipitated from HEK-293T cells using the anti-Myc antibody and analyzed by mass spectrometry. As expected, Myc-Pds5B co-immunoprecipitated

SMC3, SMC1A, RAD21, SA2, SA1, as well as Wapl and Sororin (Fig. 1A; Dataset EV1), confirming known interactions. Haspin, another Pds5B interactor (Goto et al, 2017; Zhou et al, 2017), was not detected, possibly due to its low expression in somatic cells (Higgins, 2003). These interactions were absent in the control IgG immunoprecipitation (Dataset EV2). Interestingly, Myc-Pds5B also specifically co-immunoprecipitated ATRX (Fig. 1A; Dataset EV1), a chromatin remodeling factor implicated in centromeric and telomeric cohesion through mechanisms that remain unknown (Lovejoy et al, 2020; Ramamoorthy and Smith, 2015; Ritchie et al, 2008). Immunoblot analysis following co-immunoprecipitation confirmed the specific association between Myc-Pds5B and endogenous ATRX, although this interaction appeared less pronounced than that between Myc-Pds5B and RAD21 (Fig. 1B). Reciprocal co-immunoprecipitation experiments further validated the interaction between Myc-Pds5B and ATRX-fused to GFP (ATRX-GFP) (Fig. 1C,D). In contrast, Myc-Pds5A was not noticeably associated with either endogenous ATRX or exogenously expressed ATRX-GFP in co-immunoprecipitation assays (Fig. EV1B,C).

To determine whether ATRX interacts with free Pds5B or with cohesin-bound Pds5B, we used the anti-Flag antibody to immunoprecipitate SFB-fused ATRX (SFB comprises a triple-tag: S tag, Flag tag, and a biotin-binding peptide) from HEK-293T cells. Subsequent immunoblotting revealed that SFB-ATRX specifically co-immunoprecipitated endogenous Pds5B, SMC1A, and RAD21 (Fig. EV1D), indicating that ATRX interacts with the cohesin complex in cells. To assess whether this interaction occurs indirectly through Pds5B, we immunoprecipitated stably expressed ATRX-GFP from HeLa cells. This analysis showed that ATRX-GFP specifically co-immunoprecipitated endogenous Pds5B, SMC1A, SMC3, and RAD21, with the strongest interaction observed with Pds5B (Fig. 1E). Notably, small interfering RNA (siRNA)-mediated knockdown of Pds5B abolished the association between ATRX-GFP and SMC1A, SMC3, and RAD21, suggesting that Pds5B is required for ATRX's interaction with cohesin core subunits. Consistently, when ATRX-GFP and Myc-tagged cohesin subunits were transiently co-expressed in HEK-293T cells, ATRX-GFP barely co-immunoprecipitated SMC1A, SMC3, RAD21, or SA2, likely due to the limited availability of endogenous Pds5B (Fig. 1F–I).

In summary, these data demonstrate that ATRX interacts with the cohesin complex in a Pds5B-dependent manner.

### Mapping the molecular interface between ATRX and Pds5B

We next investigated the molecular basis of the interaction between ATRX and Pds5B using either bacterially purified recombinant proteins or proteins transiently expressed in HEK-293T cells. To identify the region of Pds5B that binds ATRX, we generated three truncations spanning residues 1–300, 301–1120, and 1121–1447. Co-immunoprecipitation assays showed that ATRX-GFP co-immunoprecipitated the Flag-tagged Pds5B (1–300) fragment but not the 301–1120 or 1121–1447 fragments (Fig. 2A). Pull-down assays with the recombinant GST-fused Pds5B (1–300) fragment revealed clear interaction between GST-Pds5B (1–300) and ATRX-GFP (Fig. EV2A), indicating that residues 1–300 of Pds5B are critical for ATRX binding.

ATRX is a large protein containing multiple well-characterized domains (Fig. 2B). To further dissect the ATRX-Pds5B interaction,

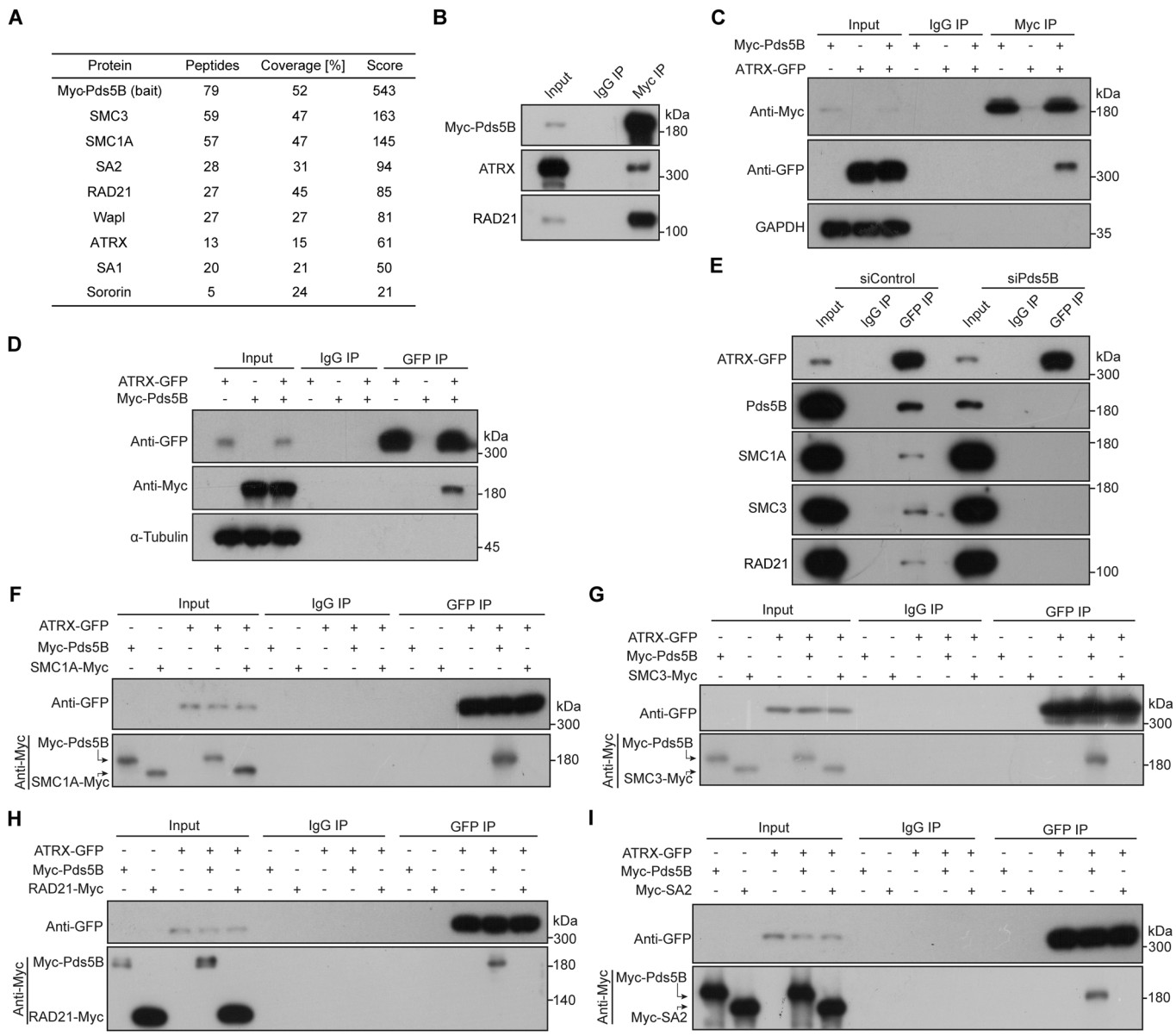

**Figure 1. A proteomic screen identifies ATRX as a Pds5B-associated factor.**

(A, B) HEK-293T cells expressing Myc-Pds5B were subjected to immunoprecipitation using anti-Myc beads or control IgG, followed by mass spectrometry (A) or immunoblotting with antibodies against the Myc tag, ATRX, and RAD21 (B). (C, D) HEK-293T cells expressing Myc-Pds5B and/or ATRX-GFP were immunoprecipitated using anti-Myc (C) or anti-GFP (D) beads, followed by immunoblotting for the Myc tag, GFP, GAPDH, or α-Tubulin as indicated. (E) HeLa cells stably expressing ATRX-GFP were transfected with control or Pds5B-targeting siRNA. Forty-eight hours post-transfection, cells were immunoprecipitated using anti-GFP beads and analyzed by immunoblotting for GFP, Pds5B, SMC1A, SMC3, and RAD21. (F–I) HEK-293T cells co-expressing ATRX-GFP and Myc-tagged SMC1A (F), SMC3 (G), RAD21 (H), or SA2 (I) were immunoprecipitated using anti-GFP beads and immunoblotted for GFP and the Myc tag. Myc-Pds5B co-transfection was used as a positive control. Source data are available online for this figure.

we examined the binding of GST-Pds5B (1–300) to various ATRX-GFP mutants (Fig. 2B). Pull-down assays showed that ATRX (846–1540)-GFP, but not ATRX (1–845)-GFP or ATRX (1541–2492)-GFP, specifically interacted with GST-Pds5B (1–300) (Fig. 2C). Further truncation of ATRX (846–1540) revealed that the ATRX (1401–1540)-GFP fragment, but not ATRX (846–1200)-GFP or ATRX (1201–1400)-GFP, bound specifically to GST-Pds5B (1–300) (Fig. 2D). Thus, the region spanning residues 1401–1540 of ATRX is required for binding to Pds5B.

To refine this interaction further, we created deletion mutants within the ATRX (1401–1540) region. Deletion of residues 1394–1443, but not 1444–1494 or 1495–1540, abolished the binding of ATRX-GFP to GST-Pds5B (1–300) (Fig. 2E), pinpointing residues 1394–1443 as crucial for the interaction. GST pull-down assays using a recombinant ATRX (1394–1443) fragment fused to GFP at the N-terminus and 2xStrep at the C-terminus showed direct interaction between GFP-ATRX (1394–1443)-2xStrep and GST-Pds5B (1–300) (Fig. EV2B), indicating that residues 1394–1443 are sufficient for Pds5B binding.

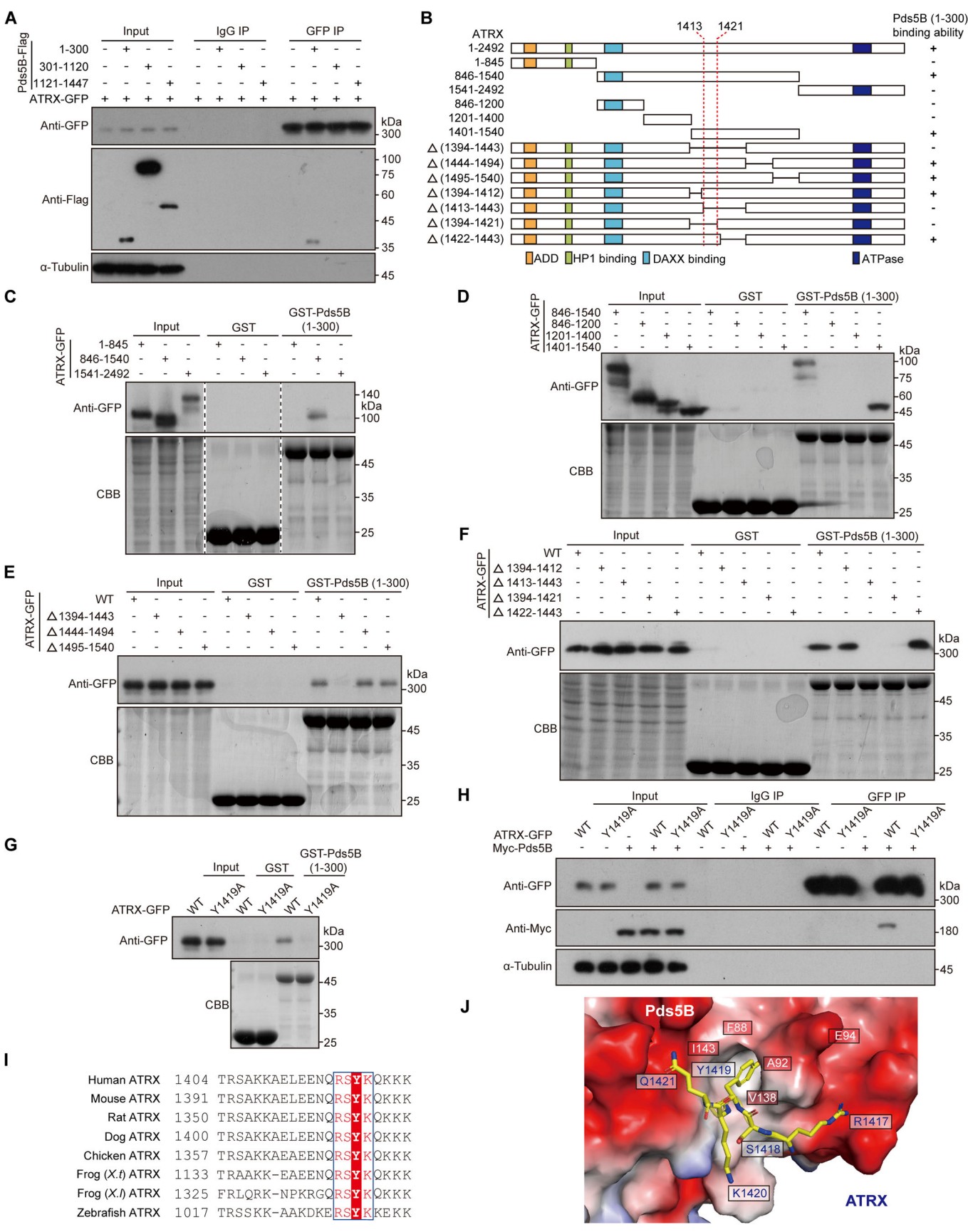

**Figure 2.  Mapping the molecular interface between ATRX and Pds5B.**

(A) HEK-293T cells co-expressing ATRX-GFP and the indicated Pds5B-Flag truncations were immunoprecipitated using anti-GFP beads and analyzed by immunoblotted for GFP, the Flag tag, and α-Tubulin. (B) Schematic representation of GST-Pds5B (1–300) binding to ATRX mutants. (C–G) Pull-down assays of the indicated ATRX-GFP truncations using GST or GST-Pds5B (1–300), followed by immunoblotting for GFP and Coomassie Brilliant Blue (CBB) staining. (H) HEK-293T cells expressing ATRX-GFP (WT or Y1419A) and/or Myc-Pds5B were immunoprecipitated using anti-GFP beads and analyzed by immunoblotting for GFP, the Myc tag, and α-Tubulin. (I) Alignment of ATRX sequences from multiple species surrounding the conserved RSYK motif. (J) AlphaFold3 model of the ATRX (RSYKQ) peptide bound to Pds5B (1–300). ATRX (RSYKQ) is shown as sticks, and Pds5B as an electrostatic surface. The interaction between Y1419 of ATRX and Pds5B residues F88, A92, V138, and I143 is highlighted. Source data are available online for this figure.

Further deletion mapping and pull-down assays showed that deletions of 1394–1421 or 1413–1443, but not 1394–1412 or 1422–1443, disrupted the binding of ATRX-GFP to GST-Pds5B (1–300) (Fig. 2F), revealing that residues 1413–1421 are essential for the interaction. Notably, systematical alanine substitutions within residues 1413–1421 of ATRX revealed that the mutation of Y1419, but not of other residues, disrupted the interaction between GFP-ATRX (1394–1443)-2xStrep and GST-Pds5B (1–300) (Fig. EV2C–E). We further observed that GFP-ATRX (1394–1443)-2xStrep, but not the Y1419A mutant, pulled down Myc-Pds5B (Fig. EV2F). Moreover, mutation of Y1419 abolished the pull-down of ATRX-GFP by GST-Pds5B (1–300) and the co-immunoprecipitation of ATRX-GFP with Myc-Pds5B (Fig. 2G,H).

Alignment of Y1419-containing short sequences of ATRX orthologs revealed that residues RSYK are conserved across vertebrates and mammals (Fig. 2I). AlphaFold3-based structural modeling predicted that Y1419 of ATRX interacts with a hydrophobic pocket of Pds5B formed by residues F88, A92, V138, and I143 (Fig. 2J), and that R1417 of ATRX forms electrostatic interaction with Q47 of Pds5B (Fig. EV2G). Consistent with this model, mutations F88A, A92P, V138A, and I143A of Pds5B, but not E94A which does not contribute to the formation of the Y1419-binding hydrophobic pocket, disrupted ATRX binding (Fig. EV2H,I). In addition, the charge-reversal mutation, R1417E, in ATRX substantially reduced Pds5B binding (Fig. EV2J).

Taken together, these data indicate that ATRX directly binds to Pds5B, and that this interaction is mediated by a conserved RSYK-[Q/K] motif in ATRX, with residue Y1419 playing a central role.

## ATRX promotes centromeric cohesion and prevents chromosome missegregation

The interaction between ATRX and Pds5B prompted us to investigate the role of ATRX in chromosome behavior during mitosis. First, we assessed the effects of ATRX knockdown, achieved by three independent siRNA duplexes (Fig. 3A), on chromosome alignment in HeLa cells. Cells were transfected with either control or ATRX-targeting siRNA and arrested in metaphase using the proteasome inhibitor MG132, which prevents anaphase onset. Knockdown of ATRX resulted in a moderate defect in chromosome alignment in cells arrested in metaphase for up to 4 h. However, after prolonged MG132 treatment (6–10 h), ATRX-depleted cells exhibited a much more severe misalignment phenotype compared to control cells (Fig. 3B,C). Specifically, after 8 h of MG132 treatment, 43.8–48.4% of ATRX-depleted cells displayed chromosome misalignment, while only 15.8% of control cells exhibited this defect.

To further assess the impact of ATRX depletion on chromosome alignment, we performed time-lapse live-cell imaging in HeLa cells stably expressing histone H2B-GFP. Cells were synchronized in monopolar mitosis using the Eg5 inhibitor S-trityl-L-cysteine (STLC) and subsequently released into fresh medium containing MG132. Control and ATRX-depleted cells completed metaphase plate formation within 75.4 and 89.8 min after STLC release, respectively (Figs. 3D and EV3A; Movies EV1 and EV2). During live imaging, 90.5% of control cells maintained chromosome alignment for up to 762 min, whereas only 47.7% of ATRX-depleted cells did so. Notably, 52.3% of ATRX-depleted cells exhibited chromosome scattering from the metaphase plate, indicating that ATRX depletion disrupts the maintenance of metaphase chromosome alignment.

The defect in maintaining chromosome alignment during prolonged metaphase suggests impaired sister chromatid cohesion (Daum et al, 2011). To investigate this, we examined the effect of ATRX depletion on sister chromatid cohesion by preparing chromosome spreads from cells arrested in metaphase with MG132 treatment. Knockdown of ATRX did not result in an obvious loss of sister chromatid cohesion in cells treated with MG132 for 2 h. Additionally, the telomeres and arms of sister chromatids were similarly resolved in both control and ATRX-depleted cells (Fig. EV3B,C). However, after 8 h of MG132 treatment, 41.8–47.5% of ATRX-depleted cells exhibited premature sister chromatid separation, compared to only 13.4% of control cells (Fig. 3E,F). These results indicate that ATRX is required to maintain sister chromatid cohesion during metaphase.

We then evaluated the impact of ATRX depletion on the strength of centromeric cohesion, by measuring the distance between sister kinetochores on chromosome spreads from HeLa cells briefly arrested in mitosis using the microtubule destabilizer nocodazole. Under these conditions, premature sister chromatid separation was rarely observed in either control or ATRX-depleted cells. However, the inter-kinetochore (inter-KT) distance increased by 13.5–15.6% in ATRX-depleted cells compared to control cells (Fig. EV3D–G). These data suggest that ATRX depletion weakens centromeric cohesion.

Weakened centromeric cohesion often leads to chromosome missegregation (Sacristan et al, 2024; Thompson et al, 2010). We then assessed the effect of ATRX depletion on chromosome segregation. Scoring anaphase cells with lagging chromosomes revealed a significant increase in the frequency of chromosome missegregation upon ATRX knockdown. In otherwise unperturbed mitosis, the percentage of anaphase cells with lagging chromosomes rose from 3.5% in control cells to 8.8–9.5% in ATRX-depleted cells (Fig. 3G,H). Additionally, following release from transient mitotic arrest induced by STLC, the frequency of anaphase cells with

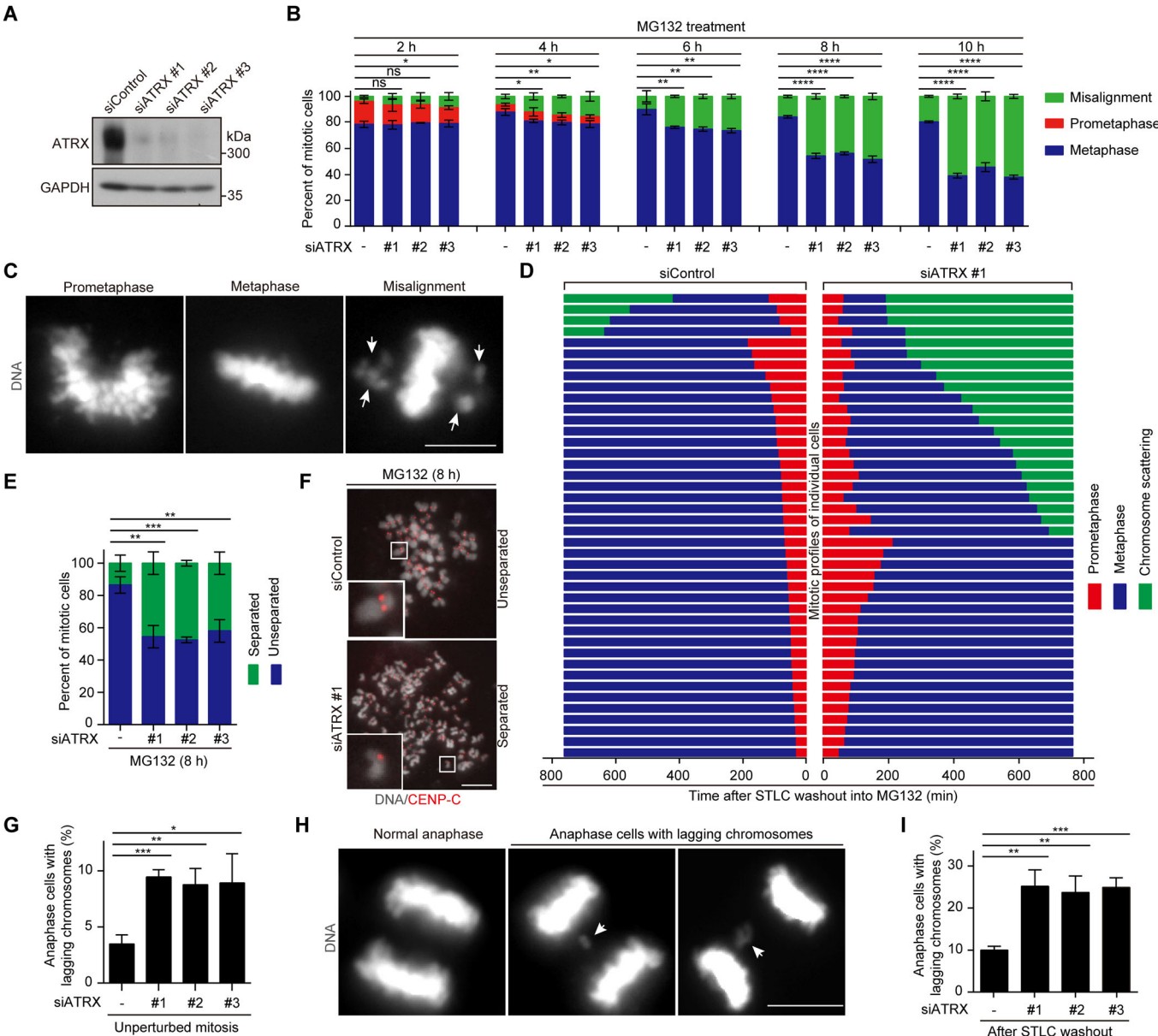

**Figure 3. ATRX promotes centromeric cohesion and prevents chromosome missegregation.**

(A–C) HeLa cells were transfected with control or ATRX siRNA duplexes. Forty-eight hours post-transfection, cells were either analyzed by immunoblotting for ATRX and GAPDH (A) or treated with MG132 and then fixed at the indicated time points for DAPI staining. The percentage of mitotic cells in prometaphase, metaphase, and pseudo-metaphase with misaligned chromosomes was quantified from more than 560 cells per condition across three independent experiments. $p$ values from left to right: ns $p = 1.04E−01$, ns $p = 0.2203$, *$p = 4.93E−02$, *$p = 4.10E−02$, **$p = 2.40E−03$, *$p = 0.018$, **$p = 5.50E−03$, **$p = 4.60E−03$, **$p = 3.60E−03$, ****$p = 3.50E−05$, ****$p = 5.00E−06$, ****$p = 3.50E−05$, ****$p = 3.00E−06$, ****$p = 6.70E−05$, ****$p = 2.00E−06$ (B). Representative images are shown, with arrows indicating misaligned chromosomes (C). (D) HeLa cells stably expressing H2B-GFP were transfected with control or ATRX siRNA and synchronized in early S phase with thymidine. Seven hours after thymidine release, cells were treated with STLC for 5 h. Mitotic cells were then collected and released into MG132-containing medium for live imaging of mitosis progression. The time from STLC washout to metaphase alignment and from metaphase to chromosome scattering or the imaging endpoint was recorded. See Movies EV1 and EV2. (E, F) HeLa cells were transfected with control or ATRX siRNAs. Forty-eight hours post-transfection, cells were treated with MG132 for 8 h. Mitotic cells were collected for chromosome spread preparation, stained with CENP-C antibodies and DAPI. The percentage of mitotic cells with predominantly separated or unseparated sister chromatids was determined from more than 930 cells per condition across three independent experiments. $p$ values from left to right: **$p = 3.04E−03$, ***$p = 4.16E−04$, **$p = 4.53E−03$ (E). Representative images are shown (F). (G, H) HeLa cells were transfected as above, then fixed and stained with DAPI. The percentage of cells displaying lagging chromosomes was quantified from more than 490 cells per condition across three independent experiments. $p$ values from left to right: ***$p = 8.13E−04$, **$p = 5.62E−03$, *$p = 2.54E−02$ (G). Representative images are shown, with arrows indicating lagging chromosomes (H). (I) HeLa cells were transfected as described above and treated with STLC for 5 h. Two hours after STLC release, cells were fixed and stained with DAPI. The percentage of anaphase cells with lagging chromosomes was quantified from more than 700 cells per condition across three independent experiments. $p$ values from left to right: **$p = 2.86E−03$, **$p = 4.19E−03$, ***$p = 4.63E−04$. Data information: Statistics were performed using unpaired Student's t-test (B, E, G, I). Means and standard deviations (SDs) are shown (B, E, G, I). ns, no significance. Scale bars, 10 μm (C, F, H). Source data are available online for this figure.

lagging chromosomes increased from 10.0% in control cells to 23.7–25.2% in ATRX-depleted cells (Fig. 3I). These findings indicate that ATRX depletion increases the frequency of erroneous chromosome segregation, suggesting impaired correction of kinetochore-microtubule attachments.

Next, we examined sister chromatid cohesion in ATRX-depleted interphase cells. We used an EGFP-fused catalytically dead version of Cas9 (dCas9) and a structurally optimized small guide (sg) RNA to label endogenous genomic loci on the arm of chromosome 15 in HeLa cells (Chen et al, 2013; Chen et al, 2018). We then measured the distance between the fluorescently labeled sister loci ("doublets") in synchronized G2-phase cells. While Sororin knockdown led to a significant increase in the distance between sister loci, depletion of ATRX did not show any measurable effect (Fig. EV3H–L). Thus, ATRX is not required for the establishment of sister chromatid cohesion during interphase.

Taken together, our data demonstrate that ATRX enhances centromeric cohesion and ensures the fidelity of chromosome segregation during mitosis.

## ATRX is enriched at mitotic centromeres in an HP1-dependent manner

Given ATRX's role in maintaining sister chromatid cohesion, we examined its localization on mitotic chromosomes. Immunofluorescence microscopy revealed ATRX enrichment at centromeres in chromosome spreads from MG132-arrested metaphase cells (Fig. 4A). Moreover, ATRX was displaced from metaphase centromeres in HeLa cells in which HP1α and HP1γ had been knocked out via CRISPR/Cas9-mediated genome editing (Fig. 4B) (Yi et al, 2018). Furthermore, ATRX and Pds5B localized exclusively to the interphase nuclei (Fig. EV4A). Additionally, ATRX colocalized with HP1α at interphase heterochromatin, and this colocalization disappeared in HP1α and HP1γ double knockout (DKO) cells (Fig. EV4B,C). These findings align with the established role of HP1-ATRX interaction in recruiting ATRX to heterochromatin (Eustermann et al, 2011; Iwase et al, 2011). Notably, ATRX's centromeric localization persisted during prolonged metaphase arrest (Fig. EV4D,E), whereas Sororin-GFP was gradually displaced from metaphase centromeres (Fig. EV4F–H).

Interestingly, super-resolution microscopy revealed that ATRX predominantly localized to the inner centromere, with 31.6% of metaphase chromosomes displaying at least two ATRX foci at bipartite centromere subdomains (Fig. 4C,D), regions that closely resemble the cohesin-accumulating sites, as reported in recent studies (Ruiz et al, 2024; Sacristan et al, 2024; Sen Gupta et al, 2023). These ATRX foci disappeared following ATRX depletion, confirming the specificity of the signal.

Next, we stably expressed siRNA-resistant ATRX-GFP, either wild-type (WT) or the Y1419A mutant forms, in HeLa cells (Fig. 4E). As expected, HP1α was co-immunoprecipitated with both ATRX-GFP and ATRX (Y1419A)-GFP, whereas Pds5B was only co-immunoprecipitated with ATRX-GFP (Fig. 4F). Super-resolution microscopy further showed that ATRX (Y1419A)-GFP predominantly localized to the inner centromeres, similar to ATRX-GFP (Fig. 4G,H). Thus, the Y1419A mutation does not impair the centromeric localization of ATRX.

Taken together, these observations indicate that ATRX is enriched at mitotic centromeres in an HP1-dependent manner.

## ATRX maintains centromeric cohesion through interaction with Pds5B

We next investigated the mechanism by which ATRX protects centromeric cohesion. Given the well-established role of ATRX in regulating gene transcription (Aguilera and López-Contreras, 2023), we first examined whether it promotes cohesion by modulating the expression of cohesin subunits. However, immunoblot analysis revealed that ATRX depletion did not detectably alter the protein levels of cohesin subunits (Fig. EV5A), thereby ruling out this possibility.

We then examined whether ATRX's protective role in centromeric cohesion depends on its interaction with Pds5B, using HeLa cells stably expressing siRNA-resistant ATRX-GFP (either WT or the Y1419A mutant). Chromosome spreads prepared from cells treated with MG132 for 8 h showed that ATRX depletion led to a pronounced loss of sister chromatid cohesion in both control HeLa cells and cells expressing ATRX (Y1419A)-GFP, but not in cells expressing ATRX-GFP (Fig. 5A–C). Further analysis of the inter-kinetochore distance on chromosome spreads from nocodazole-arrested mitotic cells showed that the centromeric cohesion defect caused by endogenous ATRX depletion was rescued by stable expression of ATRX-GFP, but not by ATRX (Y1419A)-GFP (Figs. 5D,E and EV5B,C). These findings indicate that the Y1419A mutation impairs ATRX's ability to protect centromeric cohesion. Consistently, knockdown of endogenous ATRX caused a marked defect in maintaining metaphase chromosome alignment in cells expressing ATRX (Y1419A)-GFP, but not in those expressing ATRX-GFP (Fig. 5F).

Compromised sister chromatid cohesion typically activates the spindle assembly checkpoint, leading to accumulation of cells in a prometaphase-like state. In line with this, ATRX knockdown slightly increased the mitotic index by 2.2-fold and 2.6-fold in control HeLa cells and ATRX (Y1419A)-GFP-expressing cells, respectively (Fig. EV5D,E), whereas no noticeable change was observed in ATRX-GFP-expressing cells.

Moreover, endogenous ATRX depletion significantly increased the rate of chromosome missegregation in cells expressing ATRX (Y1419A)-GFP, but not in those expressing ATRX-GFP. This was evident during unperturbed mitosis as well as after release from transient STLC treatment (Figs. 5G,H and EV5F).

Together, these results demonstrate that the interaction between ATRX and Pds5B is essential for maintaining centromeric cohesion and preventing chromosome missegregation.

## Centromere-tethering of a Pds5B-binding fragment of ATRX is sufficient to protect centromeric cohesion in ATRX-depleted cells

Building on our observations that ATRX is enriched at mitotic centromeres (Fig. 4A–C), that the ATRX (1394–1443) fragment is both necessary and sufficient for binding Pds5B (Figs. 2E and EV2B,F), and that the ATRX-Pds5B interaction is required for protecting centromeric cohesion (Fig. 5A,D), we asked whether tethering the ATRX (1394–1443) fragment to centromeres could rescue centromeric cohesion in ATRX-depleted cells.

To test this hypothesis, we stably expressed the ATRX (1394–1443) fragment (either WT or the Y1419A mutant) fused to the centromere-targeting domain of CENP-B (referred to as CB)

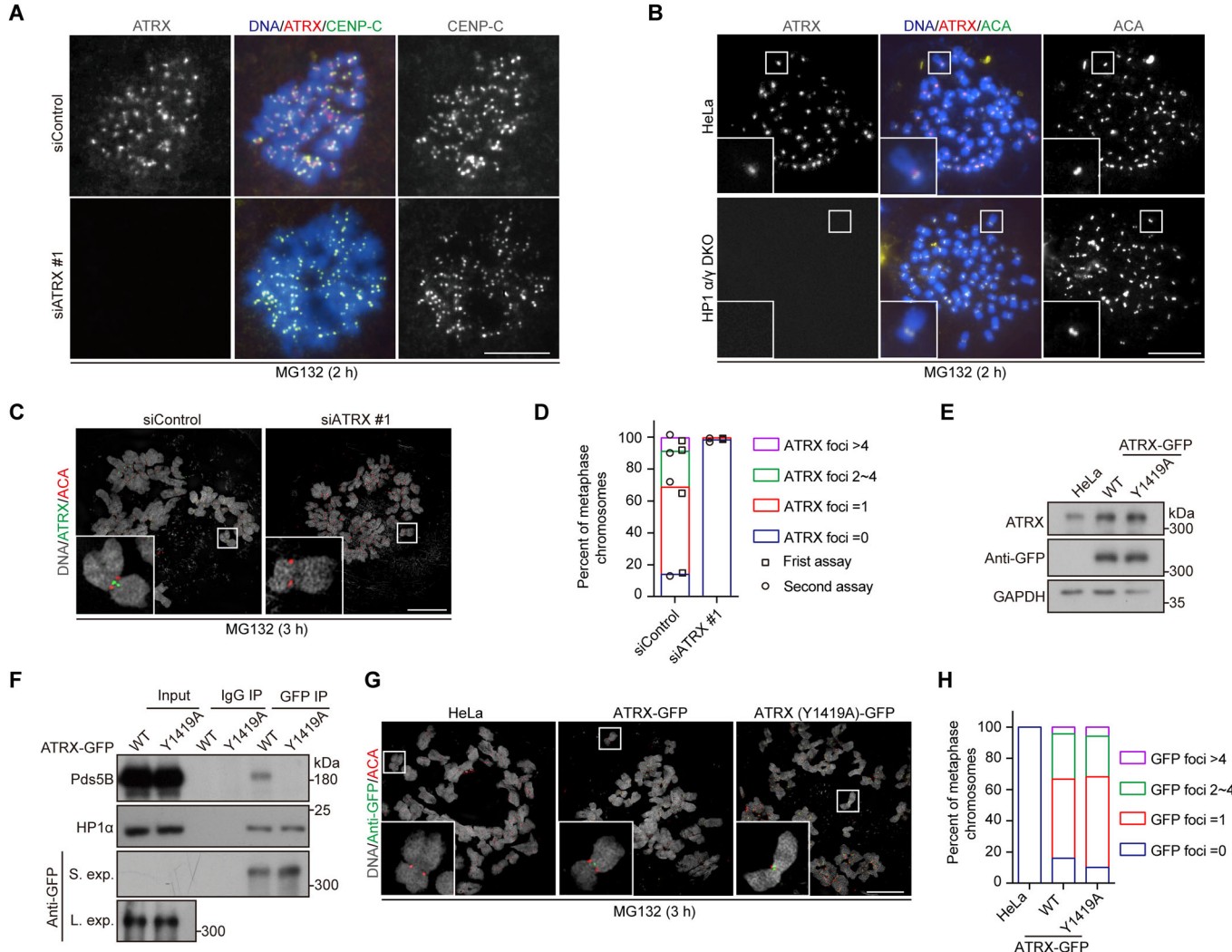

**Figure 4. ATRX is enriched at mitotic centromeres in an HP1-dependent manner.**

(A) HeLa cells were transfected with control or ATRX siRNAs. Forty-eight hours post-transfection, cells were treated with MG132 for 2 h. Mitotic cells were collected for chromosome spread preparation and subjected to immunofluorescence staining with ATRX and CENP-C antibodies, and DAPI. (B) HeLa and HP1α/γ DKO cells were treated with MG132 for 2 h, and mitotic chromosome spreads were stained for ATRX, ACA, and DAPI. (C, D) HeLa cells were transfected with control or ATRX siRNA. Forty-eight hours post-transfection, cells were treated with MG132 for 3 h, and mitotic chromosome spreads were stained for ATRX, ACA, and DAPI. Representative super-resolution images are shown (C). The number of centromeric ATRX foci was quantified from more than 299 chromosomes per condition across two independent experiments (D). (E) HeLa cells stably expressing siRNA-resistant ATRX-GFP (WT or Y1419A) were analyzed by immunoblotting for ATRX, GFP, and GAPDH. (F) HeLa cells stably expressing ATRX-GFP (WT or Y1419A) were arrested in mitosis with nocodazole, immunoprecipitated using anti-GFP beads, and analyzed by immunoblotting for Pds5B, HP1α, and GFP. S. exp., short exposure; L. exp., long exposure. (G, H) HeLa cells stably expressing ATRX-GFP (WT or Y1419A) were treated with MG132 for 3 h, and mitotic chromosome spreads were stained for GFP, ACA, and DAPI. Representative super-resolution images are shown (G). The number of centromeric ATRX foci was quantified from more than 46 chromosomes (H). Data information: Scale bars, 10 μm (A, B, C, G). Source data are available online for this figure.

in HeLa cells (Fig. 6A). Chromosome spreads from metaphase cells treated with MG132 for 8 h revealed that depletion of endogenous ATRX caused a marked loss of sister chromatid cohesion in cells expressing CB-GFP or CB-ATRX (1394–1443)-Y1419A-GFP, but not in those expressing CB-ATRX (1394–1443)-GFP (Fig. 6B–D). Immunofluorescence microscopy showed that CB-ATRX (1394–1443)-GFP only minimally recruited either Pds5B or SA2 to mitotic centromeres, in clear contrast to the robust recruitment observed with CB-fused RAD21-GFP (Appendix Fig. S1A,B). These observations suggest that CB-ATRX (1394–1443)-GFP does not promote cohesion by actively recruiting Pds5B or cohesin to

centromeres. Instead, it likely protects centromeric cohesion by binding to pre-existing, Pds5B-associated cohesin complexes at mitotic centromeres. Notably, expression of untethered ATRX (1394–1443)-GFP failed to rescue the cohesion defect following ATRX knockdown (Appendix Fig. S1C–E). Thus, centromere-tethering of the Pds5B-binding competent ATRX (1394–1443) fragment is sufficient to maintain sister chromatid cohesion in the absence of endogenous ATRX.

Further analysis of the inter-kinetochore distance showed that ATRX depletion increased the distance by 15.0, 5.3, and 16.2% in cells expressing CB-GFP, CB-ATRX (1394–1443)-GFP, and

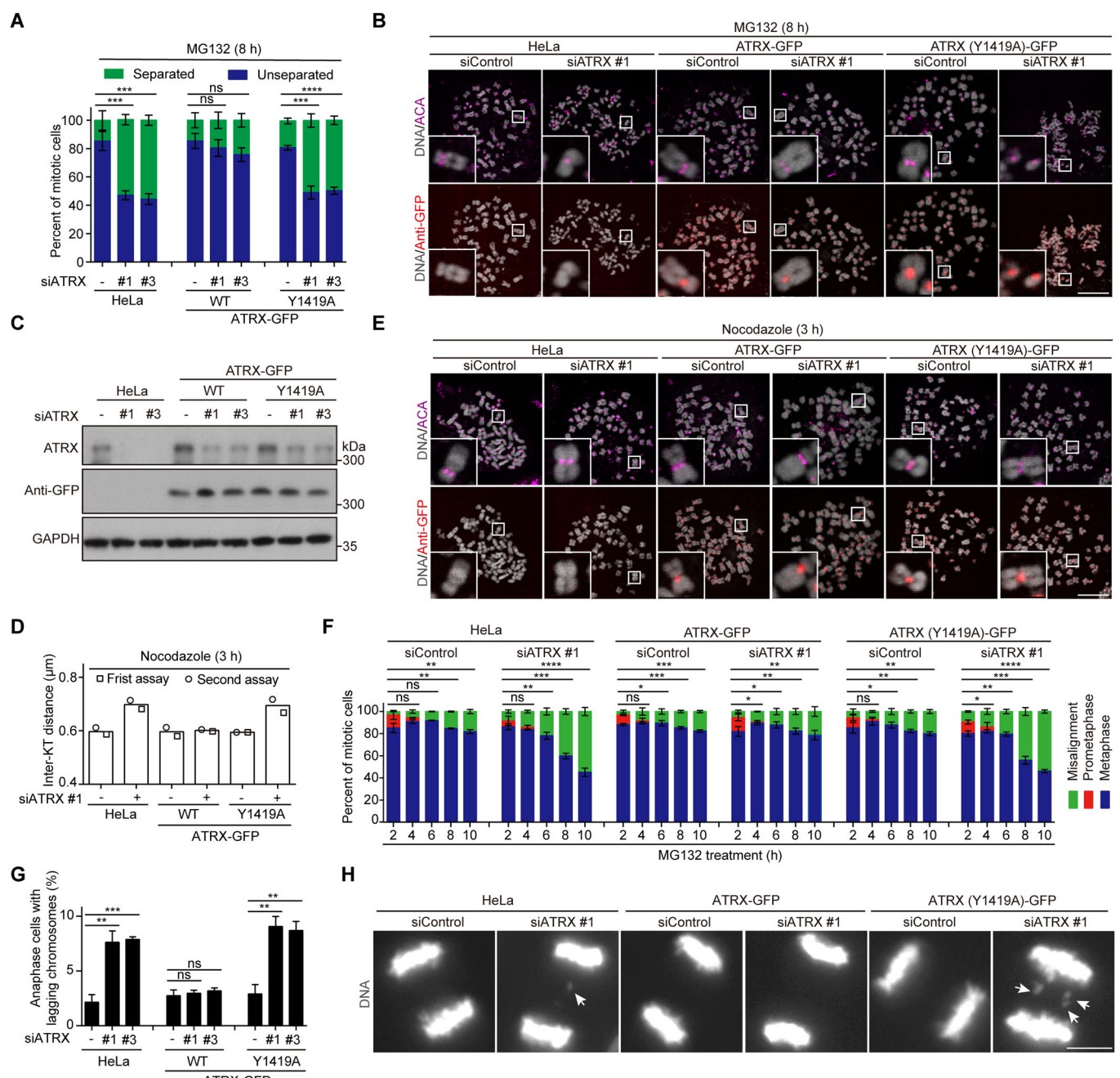

CB-ATRX (1394–1443)-Y1419A-GFP, respectively (Fig. 6E–H). This indicates that tethering the Pds5B-binding ATRX (1394–1443) fragment to centromeres bypasses the requirement for endogenous ATRX in maintaining centromeric cohesion. Consistently, only cells expressing CB-ATRX (1394–1443)-GFP, but not CB-GFP or CB-ATRX (1394–1443)-Y1419A-GFP, maintained proper metaphase chromosome alignment despite ATRX depletion (Fig. 6I). Moreover, in otherwise unperturbed mitosis, ATRX depletion increased the rate of chromosome missegregation by 4.2-, 1.5-, and 4.4-fold in cells expressing CB-GFP, CB-ATRX (1394–1443)-GFP, and CB-ATRX (1394–1443)-Y1419A-GFP, respectively (Fig. 6J).

Together, these findings demonstrate that tethering a Pds5B-binding fragment of ATRX to the centromere can circumvent the

need for endogenous ATRX in protecting centromeric cohesion, and strongly suggest that ATRX regulates sister chromatid cohesion independently of its chromatin remodeling activity.

## Wapl depletion bypasses the requirement for ATRX in maintaining centromeric cohesion

During normal mitosis, Wapl releases the majority of cohesin from chromosome arms, while a small pool of cohesin is protected from Wapl at the centromeres (Haarhuis et al, 2014). This led us to hypothesize that Wapl may be aberrantly activated at centromeres in cells lacking the ATRX-Pds5B interaction, thereby weakening centromeric cohesion. If this is the case, Wapl depletion would be

**Figure 5. ATRX maintains centromeric cohesion through interaction with Pds5B.**

(A–C) HeLa and HeLa cells stably expressing ATRX-GFP (WT or Y1419A) were transfected with control or ATRX siRNAs. Forty-eight hours post-transfection, cells were treated with MG132 for 8 h, and mitotic chromosome spreads were stained for GFP, ACA, and DAPI. The percentage of mitotic cells exhibiting predominantly separated or unseparated sister chromatids was quantified from more than 500 cells per condition across three independent experiments. p values from left to right: ***$p = 8.95E-04$, ***$p = 7.60E-04$, ns $p = 3.31E-01$, ns $p = 7.75E-02$, ***$p = 3.77E-04$, ****$p = 9.60E-05$ (A). Representative images are shown (B). Cell lysates were immunoblotted for ATRX, GFP, and GAPDH (C). (D, E) Cells transfected with siRNAs as described above were treated with nocodazole for 3 h, and mitotic chromosome spreads were stained for GFP, ACA, and DAPI. The inter-KT distance was measured on more than 1750 chromosomes from 80 cells per condition across two independent experiments (see Fig. EV5B,C). Means and ranges are plotted (D). Representative images are shown (E). (F) Cells transfected with siRNAs as described above were treated with MG132 and fixed at the indicated time points for DAPI staining. The percentages of mitotic cells in prometaphase, metaphase, and pseudo-metaphase exhibiting misaligned chromosomes were determined from more than 620 cells per condition across three independent experiments. p values from left to right: ns $p = 1.40E-01$. ns $p = 6.74E-02$, **$p = 1.71E-03$, **$p = 1.48E-03$. ns $p = 6.43E-02$. **$p = 6.81E-03$, ***$p = 4.86E-04$, ****$p = 4.70E-05$ (HeLa). ns $p = 5.39E-02$. *$p = 1.32E-02$, ***$p = 6.11E-04$, ***$p = 5.09E-04$. *$p = 2.47E-02$. *$p = 2.26E-02$, **$p = 1.60E-03$, **$p = 1.30E-03$ (ATRX-GFP). ns $p = 1.47E-01$. *$p = 1.99E-02$, **$p = 1.64E-03$, **$p = 1.25E-03$. *$p = 4.43E-02$. **$p = 2.78E-03$, ***$p = 1.38E-04$, ****$p = 4.25E-07$ (ATRX-Y1419A-GFP). (G, H) Cells transfected with siRNAs as described above were fixed and stained with DAPI. The percentage of anaphase cells exhibiting lagging chromosomes was determined from more than 530 cells per condition across three independent experiments. p values from left to right: **$p = 1.69E-03$, ***$p = 1.86E-04$, ns $p = 5.64E-01$, ns $p = 2.70E-01$, **$p = 1.06E-03$, **$p = 1.16E-03$ (G). Representative images are shown, with arrows indicating lagging chromosomes (H). Data information: Statistics were performed using unpaired Student's t-test (A, F, G). Means and SDs are shown (A, F, G). ns, no significance. Scale bars, 10 μm (B, E, H). Source data are available online for this figure.

expected to restore proper centromeric cohesion in the absence of ATRX.

To test this, we examined sister chromatid cohesion in HeLa cells depleted of Wapl and/or ATRX. After treatment with MG132 for 8 h, 14.7, 40.1, 10.0, and 13.2% of cells displayed premature sister chromatid separation following transfection with control siRNA, ATRX siRNA, Wapl siRNA, and a combination of ATRX and Wapl siRNAs, respectively (Fig. 7A–C). Further analysis of chromosome spreads from nocodazole-arrested mitotic cells showed that ATRX knockdown increased the inter-kinetochore distance by 11.2% in the presence of Wapl, but only by 1.7% in its absence (Fig. 7D–G). Immunofluorescence microscopy revealed that ATRX remained enriched at metaphase centromeres in Wapl-depleted cells (Fig. 7H,I), consistent with its HP1-dependent localization at mitotic centromeres (Fig. 4B). Thus, ATRX is not required for maintaining centromeric cohesion in the absence of Wapl.

We next assessed the impact of ATRX depletion on metaphase chromosome alignment in the presence or absence of Wapl. While ATRX depletion alone caused a severe defect in chromosome alignment during metaphase arrest, cells depleted of Wapl alone or co-depleted of both Wapl and ATRX maintained proper chromosome alignment (Fig. 7J). Thus, ATRX is not required for maintaining metaphase chromosome alignment when Wapl is absent.

Taken together, these results indicate that the role of ATRX in protecting centromeric cohesion can be bypassed by Wapl depletion.

## ATRX antagonizes the direct interaction between Wapl and Pds5B

The interaction between Wapl and Pds5 facilitates Wapl's activity in releasing cohesin (Liang et al, 2018; Ouyang et al, 2016; Shintomi and Hirano, 2009). Given our findings that the interaction with Pds5B is required for ATRX to protect centromeric cohesion (Fig. 5A,D), that tethering a Pds5B-binding fragment of ATRX to centromeres is sufficient to protect sister chromatid cohesion (Fig. 6B,E), and that Wapl depletion bypasses the requirement for ATRX in maintaining centromeric cohesion (Fig. 7A,D), we hypothesized that the ATRX-Pds5B interaction may antagonize the binding of Wapl to Pds5B.

The YSR motif is essential for Wapl to bind Pds5B (Ouyang et al, 2016). GST pull-down assays confirmed that a short Wapl fragment (residues 1–28) containing the YSR motif, but not a mutant in which YSR was replaced by ASE, effectively bound to Myc-Pds5B (Fig. 8A). Sequence alignment revealed that the Y1419-containing RSYK-[Q/K] motif of ATRX shares similarity with the YSR-containing KTYSR motif of Wapl (Fig. 8B), suggesting that ATRX and Wapl may bind overlapping sites on Pds5B. Indeed, mutations at Pds5B residues F88 and A92, but not E94, disrupted binding of GST-Pds5B (1–300) to Wapl-GFP (Fig. 8C).

To test whether ATRX and Wapl compete for Pds5B binding, we leveraged our observation that recombinant GFP-ATRX (1394–1443)-2xStrep binds Pds5B (Fig. EV2B,F). Strikingly, GFP-ATRX (1394–1443)-2xStrep, but not the Y1419A mutant, markedly reduced the binding of Wapl (1–28)-GST to Myc-Pds5B in HEK-293T cells (Fig. 8D), and to recombinant MBP-Pds5B (1–300) (Fig. 8E). Similarly, in HeLa cells stably expressing Myc-Pds5B, GFP-ATRX (1394–1443)-2xStrep, but not the Y1419A mutant, inhibited co-immunoprecipitation of endogenous Wapl with Myc-Pds5B in a dose-dependent manner (Fig. 8F). Thus, the Pds5B-binding fragment of ATRX can directly antagonize the YSR motif-dependent interaction between Wapl and Pds5B.

We next asked whether ATRX and Wapl bind Pds5B mutually exclusively in cells. In HeLa cell lysates, ATRX co-immunoprecipitated Pds5B but not Wapl (Fig. 8G), while stably expressed Wapl-GFP co-immunoprecipitated Pds5B but not ATRX (Fig. 8H). Moreover, stably expressed Myc-Pds5B co-immunoprecipitated both Wapl and ATRX in cells synchronized in either G2 phase or mitosis (Fig. 8I). We also noticed that Myc-Pds5B co-immunoprecipitated slightly more ATRX in mitotic cells than in G2-phase cells. In contrast, Myc-Pds5B co-immunoprecipitated Sororin from G2-phase cells but not from mitotic cells, consistent with the phosphorylation-dependent disruption of the Sororin-Pds5B interaction during mitosis (Nishiyama et al, 2010).

We further observed that ATRX protein levels gradually increased from G1 to S phase (Appendix Fig. S2A), and that Myc-Pds5B co-immunoprecipitated Wapl much more efficiently than ATRX in G1-phase cells (Appendix Fig. S2B). Additionally, ATRX knockdown did not affect the interaction between Myc-Pds5B and Wapl in G1-phase cells (Appendix Fig. S2B), nor the binding of Myc-Pds5B to Wapl and Sororin in asynchronous cells (Appendix Fig. S2C). These observations suggest that ATRX does not interfere with Pds5B binding to Wapl and Sororin in interphase cells.

In summary, our results demonstrate that ATRX binding to Pds5B antagonizes the interaction between Wapl and Pds5B, revealing a competitive mechanism that safeguards centromeric cohesin.

## ATRX protects sister chromatid cohesion independently of Haspin

Our results demonstrate that ATRX strengthens centromeric cohesion by binding to Pds5B through a conserved RSYK-[Q/K] motif. Previous studies have reported that Haspin promotes centromeric cohesion by interacting with Pds5B via a similar [R/K]-TYG-[R/K] motif (Goto et al, 2017; Zhou et al, 2017). We

next assessed the effect of ATRX knockdown on sister chromatid cohesion in HeLa cells with CRISPR/Cas9-mediated Haspin knockout (KO).

Consistent with our previous findings (Zhou et al, 2017), Haspin KO cells exhibited defects in maintaining sister chromatid cohesion during metaphase arrest induced by MG132 treatment for 2–6 h (Fig. 9A–C). Notably, ATRX knockdown further exacerbated cohesion loss in Haspin KO cells, indicating that ATRX contributes to sister chromatid cohesion independently of Haspin. ATRX depletion also aggravated metaphase chromosome alignment defects in Haspin KO cells (Fig. 9D). Moreover, in otherwise unperturbed mitotic Haspin KO cells, the frequency of anaphases with lagging chromosomes increased from 10.3% following control

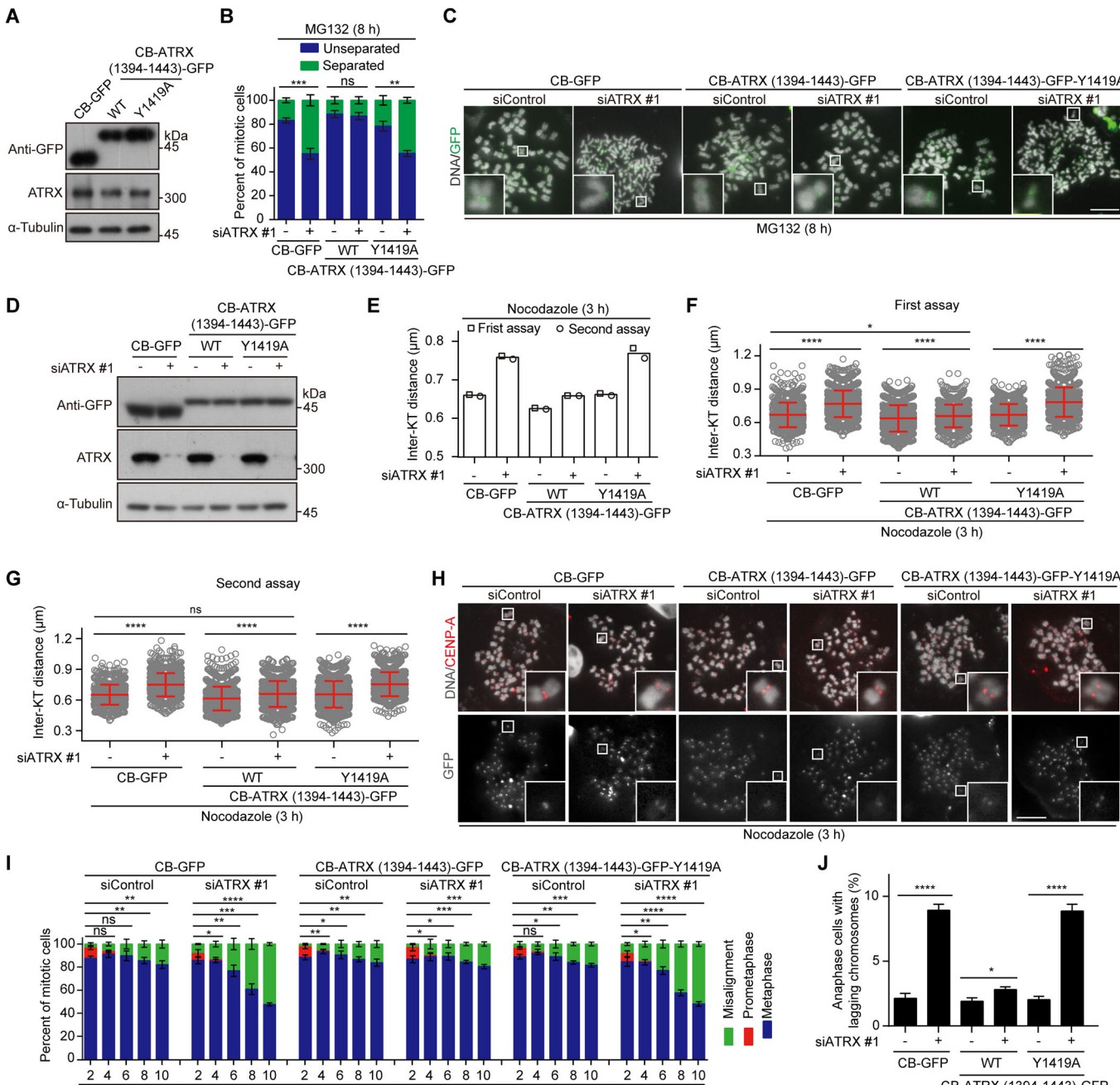

**Figure 6. Centromere-tethering of a Pds5B-binding fragment of ATRX is sufficient to protect centromeric cohesion in ATRX-depleted cells.**

(A) HeLa cells stably expressing CB-GFP or CB-ATRX (1394–1443)-GFP (WT or Y1419A) were analyzed by immunoblotting for GFP, ATRX, and α-Tubulin. (B–D) HeLa cells stably expressing the indicated proteins were transfected with control or ATRX siRNA. Forty-eight hours post-transfection, cells were treated with MG132 for 8 h, and mitotic chromosome spreads were stained with DAPI. The percentage of mitotic cells exhibiting predominantly separated or unseparated sister chromatids was quantified from more than 770 cells per condition across three independent experiments. $p$ values from left to right: ***$p = 6.69E−04$, ns $p = 5.30E−01$, **$p = 1.04E−03$ (B). Representative images are shown (C). Cell lysates were analyzed by immunoblotting for GFP, ATRX, and α-Tubulin (D). (E–H) Cells transfected as described above were treated with nocodazole for 3 h, and mitotic chromosome spreads were stained for CENP-A and DAPI. The inter-KT distance was measured on more than 2000 chromosomes from 90 cells per condition across two independent experiments, and the means and ranges are plotted (E). Inter-KT distances from two individual experiments are shown. $p$ values from left to right: ****$p < 1.00E−15$, *$p = 3.62E−02$, ****$p = 7.27E−06$ (F). ****$p < 1.00E−15$, ns $p = 1.11E−01$, ****$p < 1.00E−15$ (G). Representative images are shown (H). (I) Cells transfected as described above were treated with MG132 and fixed at the indicated time points for DAPI staining. The percentages of mitotic cells in prometaphase, metaphase, and pseudo-metaphase with misaligned chromosomes were determined from more than 670 cells per condition across three independent experiments. $p$ values from left to right: ns $p = 6.74E−02$. ns $p = 6.02E−02$, **$p = 3.47E−03$, **$p = 2.39E−03$. *$p = 2.21E−02$. **$p = 6.22E−03$, ***$p = 3.34E−04$, ****$p = 6.14E−07$ (CB-GFP). **$p = 1.20E−03$. *$p = 1.1E−02$, **$p = 1.21E−03$, **$p = 1.55E−03$. *$p = 4.76E−02$. *$p = 1.50E−02$, ***$p = 1.25E−04$, ***$p = 1.10E−04$ (CB-ATRX (1394–1443)-GFP). ns $p = 1.23E−01$. *$p = 3.33E−02$, **$p = 1.15E−03$, ***$p = 6.70E−04$. *$p = 1.36E−02$. **$p = 2.20E−03$, ****$p = 3.70E−05$, ****$p = 6.32E−06$ (CB-ATRX (1394–1443)-GFP-Y1419A). (J) Cells transfected as described above were fixed and stained with DAPI. The percentage of anaphase cells exhibiting lagging chromosomes was quantified from more than 650 cells per condition across three independent experiments. $p$ values from left to right: ****$p = 4.40E−05$, *$p = 1.10E−02$, ****$p = 3.90E−05$. Data information: Statistics were performed using unpaired Student's t-test (B, F, G, I, J). Means and SDs are shown (B, F, G, I, J). ns, no significance. Scale bars, 10 μm (C, H). Source data are available online for this figure.

siRNA transfection to 16.4–17.0% after ATRX depletion (Fig. 9E). Thus, ATRX and Haspin play additive roles in protecting centromeric cohesion.

Given the similarity in their Pds5B-binding motifs, we further examined the effect of centromere tethering of the Pds5B-binding fragment of ATRX in Haspin KO cells. Interestingly, expression of CB-ATRX (1394–1443)-GFP, but not CB-GFP or CB-ATRX (1394–1443)-Y1419A-GFP, mitigated the loss of metaphase sister chromatid cohesion in Haspin KO cells (Appendix Fig. S3A–C). This suggests that ATRX and Haspin promote centromeric cohesion through at least partially similar mechanisms.

Sgo1 is a well-established protector of sister chromatid cohesion (Kitajima et al, 2004; McGuinness et al, 2005; Salic et al, 2004; Tang et al, 2004). During early mitosis, Sgo1 localizes along chromosome arms (Chu et al, 2020; Gimenez-Abian et al, 2004; Nakajima et al, 2007; Yan et al, 2024), and becomes enriched at centromeres (Kitajima et al, 2005; McGuinness et al, 2005). We next investigated whether ATRX contributes to cohesin protection in the absence of Sgo1. As expected, Sgo1 knockdown led to a strong loss of sister chromatid cohesion in nocodazole-arrested mitotic cells, which was more severe than the defect caused by ATRX depletion alone (Appendix Fig. S3D–F), indicating that Sgo1 plays the dominant role, particularly during early mitosis prior to metaphase. Moreover, combined depletion of ATRX and Sgo1 resulted in an additive cohesion defect, highlighting their non-redundant contributions. Finally, we assessed whether centromere tethering of the Pds5B-binding ATRX fragment could compensate for the loss of Sgo1. Expression of CB-ATRX (1394–1443)-GFP did not noticeably rescue the cohesion defect in Sgo1-depleted cells (Appendix Fig. S3G–I), consistent with the distinct mechanisms by which ATRX and Sgo1 interact with, and protect, the cohesin complex.

## Discussion

In this study, we identify the chromatin remodeler ATRX as a novel binding partner of the cohesin complex through a proteomic screen. Our findings demonstrate that ATRX directly interacts with Pds5B to safeguard centromeric cohesion and ensure faithful chromosome segregation during mitosis. Mechanistically, we show that ATRX and Wapl compete for binding to Pds5B, with ATRX acting as an antagonist to prevent Wapl-mediated release of cohesin from mitotic centromeres (Fig. 9F). This study uncovers a previously unrecognized layer of regulation critical for chromosomal stability.

Our data show that disruption of the ATRX-Pds5B interaction leads to an increased frequency of anaphase lagging chromosomes, which is indicative of merotelic attachment between kinetochores and microtubules (Gregan et al, 2011). Interestingly, cohesin was recently shown to stabilize the bipartite subdomains of centromeric chromatin during mitosis, thereby preventing the formation of such merotelic attachments (Sacristan et al, 2024). Using super-resolution microscopy, we detected ATRX at these inner-centromere subdomains. We propose that ATRX-bound cohesin contributes to stabilizing these subdomains, thereby helping to prevent chromosome missegregation.

Through structural and mutational analyses, we discovered that ATRX binds Pds5B via a conserved RSYK-[Q/K] motif, with Y1419 serving as a key residue. This residue inserts into a hydrophobic pocket on Pds5B, which is also recognized by the YSR motif of Wapl (Ouyang et al, 2016). Although the N-terminal YSR motif may not be essential for the cohesin release activity of Wapl (Nasmyth et al, 2023), this motif clearly promotes Wapl-mediated resolution of chromosome arms in early mitosis (Ouyang et al, 2016). Among known Pds5 interactors, ATRX, Wapl, Sororin, and Haspin share a similar five-residue sequence, [R/K]-[S/T]-Y-[K/S/T/G/A]-[Q/R/K/A], which is conserved in vertebrates (Goto et al, 2017; Ouyang et al, 2016; Zhou et al, 2017). An exception is PD-L1, which binds to Pds5B through a short peptide sequence, with Y123 being the only conserved residue across vertebrates (see Fig. 8B) (Yu et al, 2020), highlighting the critical role of this residue in binding Pds5B. Our discovery of the ATRX-Pds5B interaction may also explain the interesting observation that deletion of the YSR motif-containing residues KVRRSYSRL of Sororin does not strongly affect cohesion (Wu et al, 2011). We propose that at least two functional pools of cohesin exist at mitotic centromeres: one stabilized by ATRX-bound Pds5B and another regulated by Haspin-bound Pds5B (Fig. 9F). This partitioning of regulatory factors may provide redundancy to ensure robust centromeric cohesion under varying cellular conditions.

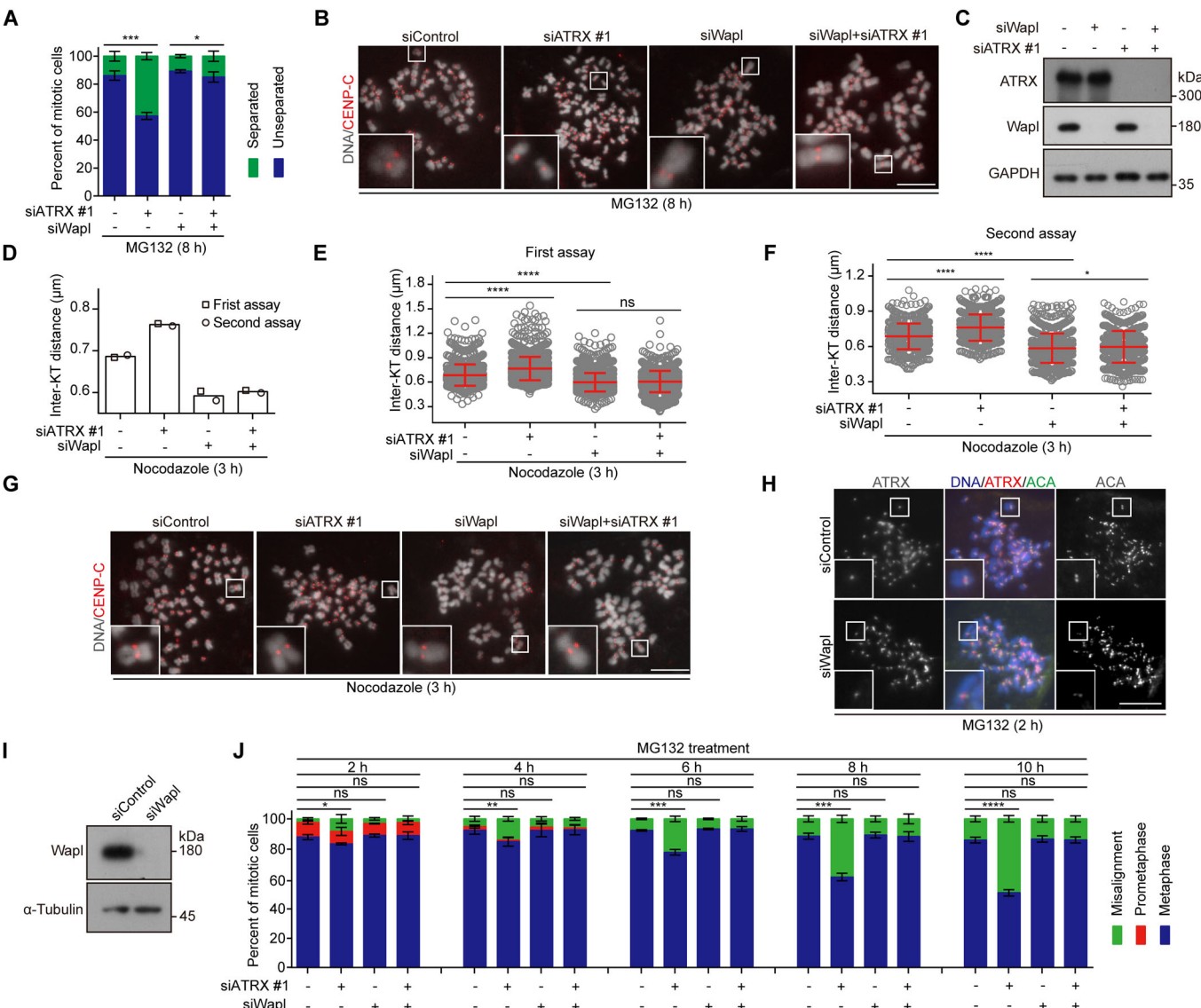

**Figure 7. Wapl depletion bypasses the requirement for ATRX in maintaining centromeric cohesion.**

(A–C) HeLa cells were transfected with control siRNA, ATRX siRNA, and/or Wapl siRNA. Forty-eight hours post-transfection, cells were treated with MG132 for 8 h, and mitotic chromosome spreads were stained for CENP-C and DAPI. The percentage of mitotic cells exhibiting predominantly separated or unseparated sister chromatids was quantified from 1000 cells per condition across three independent experiments. p values from left to right: ***p = 1.05E−04, *p = 4.10E−02 (A). Representative images are shown (B). Cell lysates were analyzed by immunoblotting for ATRX, Wapl, and GAPDH (C). (D–G) HeLa cells transfected as described above were treated with nocodazole for 3 h, and mitotic chromosome spreads were stained for CENP-C and DAPI. The inter-KT distance was measured on more than 2000 chromosomes from 95 cells per condition across two independent experiments, and the means and ranges are plotted (D). Inter-KT distances from two individual experiments are shown. p values from left to right: ****p < 1.00E−15, ****p < 1.00E−15, ns p = 8.92E−02 (E). ****p < 1.00E−15, ****p < 1.00E−15, ns p = 4.73E−02 (F). Representative images are shown (G). (H, I) HeLa cells transfected as described above were treated with MG132 for 2 h, and mitotic chromosome spreads were stained for ATRX, ACA, and DAPI. Representative images are shown (H). Cell lysates were analyzed by immunoblotting for Wapl and α-Tubulin (I). (J) HeLa cells transfected as described above were treated with MG132 and fixed at the indicated time points for DAPI staining. The percentages of mitotic cells in prometaphase, metaphase, and pseudo-metaphase exhibiting misaligned chromosomes were determined from more than 700 cells per condition across three independent experiments. p values from left to right: *p = 2.73E−02, ns p = 3.80E−01, ns p = 9.86E−01, **p = 2.36E−03, ns p = 7.80E−01, ns p = 4.34E−01, ***p = 1.85E−04, ns p = 7.17E−02, ns p = 3.57E−01, ***p = 1.21E−04, ns p = 6.58E−01, ns p = 9.20E−01, ****p = 2.90E−05, ns p = 6.78E−01, ns p = 9.71E−01. Data information: Statistics were performed using unpaired Student's t-test (A, E, F, J). Means and SDs are shown (A, E, F, J). ns, no significance. Scale bars, 10 μm (B, G, H). Source data are available online for this figure.

While ATRX contributes to the stabilization of centromeric cohesion, its depletion results in a milder cohesion defect compared to Sgo1 depletion. This is not unexpected, as ATRX specifically reinforces centromeric cohesion, whereas Sgo1 protects cohesin both on chromosome arms and at centromeres. Our data further

suggest that ATRX may become particularly important for maintaining centromeric cohesion during metaphase, after the Sgo1-PP2A complex redistributes toward kinetochore-proximal regions, a step believed to expose cohesin at the inner centromere to Wapl-mediated release. Future work will be required to dissect

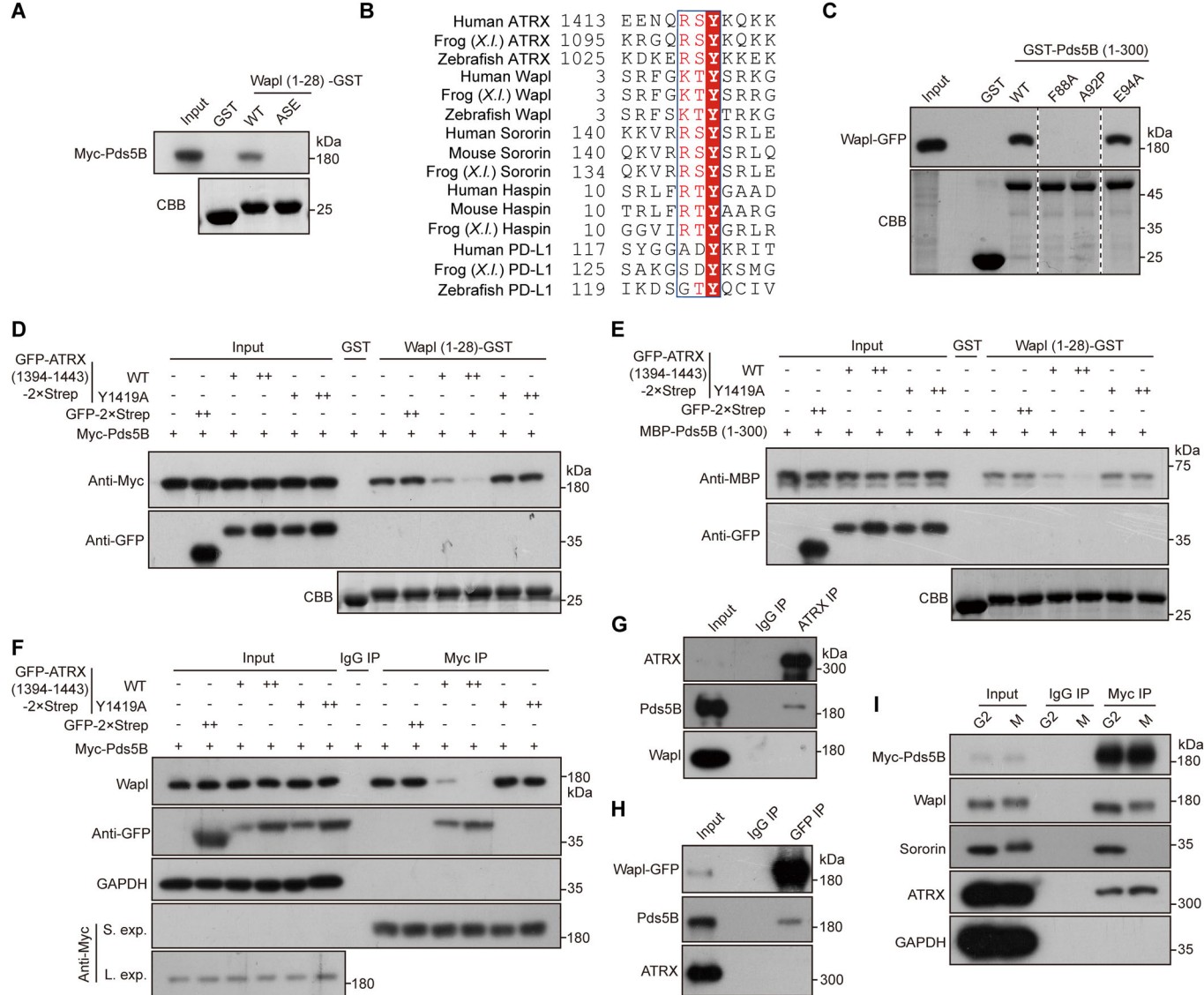

**Figure 8. ATRX antagonizes the direct interaction between Wapl and Pds5B.**

(A) HEK-293T cells expressing Myc-Pds5B were subjected to pull-down with GST or Wapl (1–28)-GST, followed by immunoblotting for the Myc tag and CBB staining. (B) Multiple sequence alignment of ATRX, Wapl, Sororin, Haspin, and PD-L1. (C) HEK-293T cells expressing Myc-Pds5B were subjected to pull-down with GST or GST-Pds5B (1–300) (WT or mutants), followed by immunoblotting for GFP and CBB staining. (D) HEK-293T cells expressing Myc-Pds5B were subjected to pull-down with GST or Wapl (1–28)-GST in the presence of GFP-2xStrep or increasing amounts of GFP-ATRX (1394–1443)-2xStrep (WT or Y1419A), followed by immunoblotting for the Myc tag, GFP, and CBB staining. (E) MBP-Pds5B (1–300) was subjected to pull-down with GST or Wapl (1–28)-GST in the presence of GFP-2xStrep or increasing amounts of GFP-ATRX (1394–1443)-2xStrep (WT or Y1419A), followed by immunoblotting for MBP, GFP, and CBB staining. (F) HeLa cells stably expressing Myc-Pds5B were immunoprecipitated using anti-Myc beads or control IgG in the presence of GFP-2xStrep or increasing amounts of GFP-ATRX (1394–1443)-2xStrep (WT or Y1419A), followed by immunoblotting for Wapl, GFP, GAPDH, and the Myc tag. S. exp., short exposure; L. exp., long exposure. (G) Nocodazole-arrested mitotic HeLa cells were immunoprecipitated using the ATRX antibody or control IgG, followed by immunoblotting for ATRX, Pds5B, and Wapl. (H) Nocodazole-arrested mitotic HeLa cells stably expressing Wapl-GFP were immunoprecipitated using anti-GFP beads or control IgG, followed by immunoblotting for GFP, Pds5B, and ATRX. (I) HeLa cells stably expressing Myc-Pds5B, synchronized in G2-phase by double thymidine block/release or arrested in mitosis by nocodazole, were immunoprecipitated using anti-Myc beads or control IgG, followed by immunoblotting for the Myc tag, Wapl, Sororin, ATRX, and GAPDH. Source data are available online for this figure.

this temporal regulation and fully elucidate the interplay between Sgo1-PP2A, ATRX, and Wapl in the protection of centromeric cohesion.

We noticed that, in co-immunoprecipitation assays using HeLa cell lysates, ATRX associated with Pds5B but was not detectably bound to Pds5A (Fig. EV1B,C), suggesting that ATRX preferentially interacts with Pds5B. This preference may be due to

differences in intracellular localization, post-translational modifications, or the distinct unstructured C-terminal domains of Pds5B and Pds5A (Carretero et al, 2013; Hellmuth and Stemmann, 2024; Losada et al, 2005). The preferential binding of Pds5B to ATRX may also reflect the specific role for Pds5B, but not Pds5A, in protecting centromeric cohesion (Carretero et al, 2013). Future studies will be needed to clarify these possibilities.

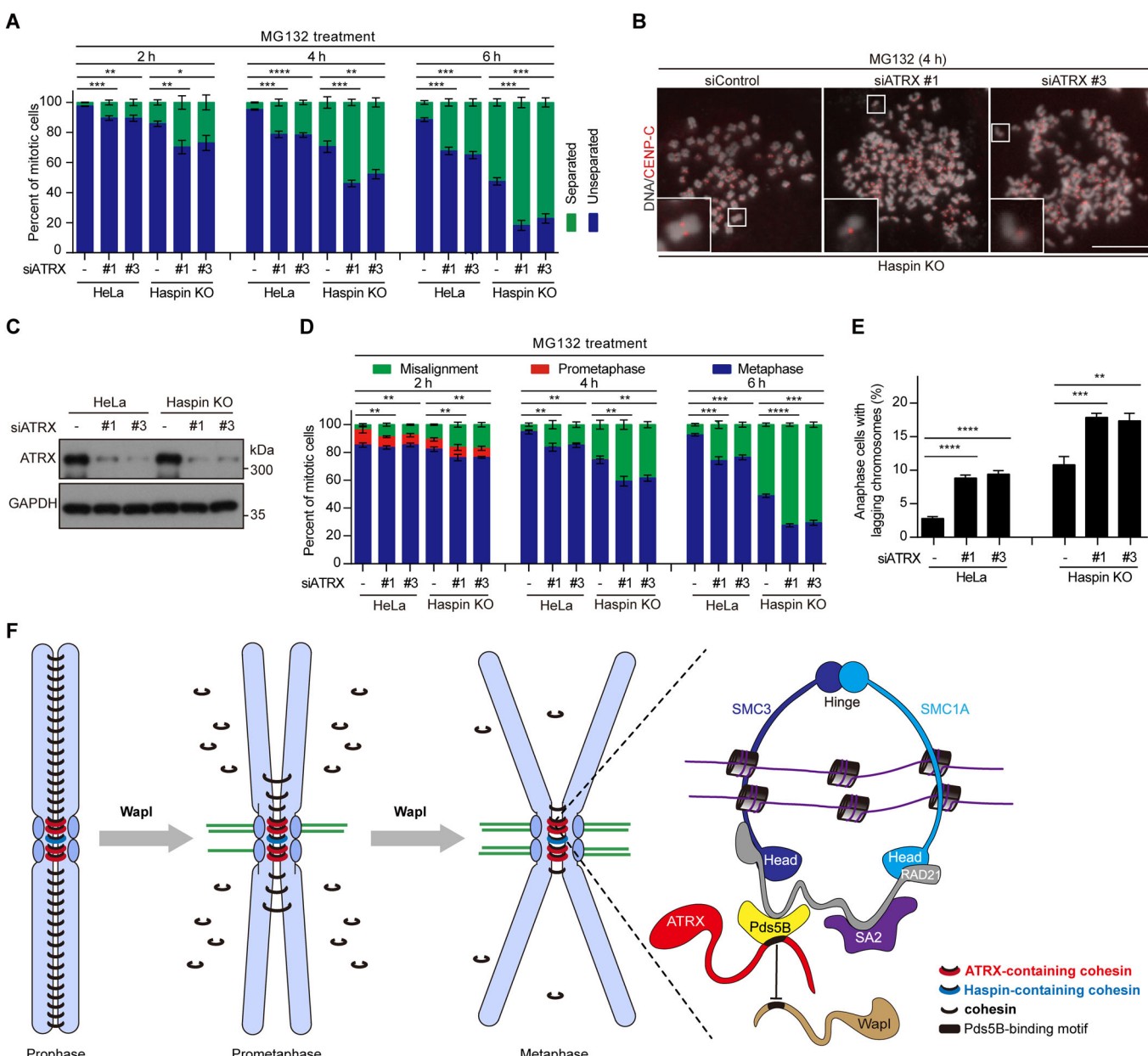

**Figure 9. ATRX protects sister chromatid cohesion independently of Haspin.**

(A–C) HeLa and Haspin KO cells were transfected with control or ATRX siRNA. Forty-eight hours post-transfection, cells were treated with MG132 for 2, 4, and 6 h, and mitotic chromosome spreads were stained with DAPI. The percentage of mitotic cells with predominantly separated or unseparated sister chromatids was quantified from more than 720 cells per condition across three independent experiments. p values from left to right: ***$p = 8.95E-04$, **$p = 2.53E-03$, **$p = 4.86E-03$, *$p = 1.38E-02$, ***$p = 2.19E-04$, ****$p = 4.70E-05$, ***$p = 6.23E-04$, **$p = 2.88E-03$, ***$p = 1.93E-04$, ***$p = 1.19E-04$, ***$p = 2.55E-04$, ***$p = 4.02E-04$ (A). Representative images are shown (B). Cell lysates were analyzed by immunoblotting for ATRX and GAPDH (C). (D) Cells transfected as described above were treated with MG132 and fixed at the indicated time points for DAPI staining. The percentage of mitotic cells in prometaphase, metaphase, and pseudo-metaphase with misaligned chromosomes was quantified from more than 740 cells per condition across three independent experiments. p values from left to right: **$p = 2.41E-03$, **$p = 2.20E-03$, **$p = 4.81E-03$, **$p = 2.70E-03$, **$p = 3.79E-03$, **$p = 2.21E-03$, **$p = 4.51E-03$, **$p = 3.71E-03$, ***$p = 3.87E-04$, ***$p = 1.42E-04$, ****$p = 2.50E-05$, ***$p = 1.08E-04$. (E) Cells transfected as described above were fixed and stained with DAPI. The percentage of anaphase cells with lagging chromosomes was determined from more than 650 cells per condition across three independent experiments. p values from left to right: ****$p = 3.70E-05$, ****$p = 5.10E-05$, ***$p = 8.31E-04$, **$p = 2.42E-03$. (F) Model for the role of ATRX in antagonizing Wapl and maintaining centromeric cohesion during mitosis. ATRX is enriched at the bipartite subdomains of the inner centromere, where it binds to Pds5B to antagonize Wapl binding, thereby preventing Wapl-mediated release of centromeric cohesin. Note that another pool of cohesin, in which Pds5B binds to Haspin, is also present at the inner centromere. Data information: Statistics were performed using unpaired Student's t-test (A, D, E). Means and SDs are shown (A, D, E). Scale bars, 10 μm (B). Source data are available online for this figure.

ATRX is best known for its roles in chromatin remodeling and transcriptional regulation (Aguilera and López-Contreras, 2023). Previous studies have suggested that the ATPase activity of an ISWI-containing chromatin remodeling complex is involved in loading cohesin onto chromosomes (Hakimi et al, 2002), and that the ISW1a chromatin remodeler in budding yeast modulates cohesin distribution in centromeric regions (Litwin et al, 2023). Moreover, the RSC chromatin remodeler generates nucleosome-free DNA templates to promote cohesin loading (Muñoz et al, 2019). However, our findings show that ATRX's role in centromeric cohesion is independent of its chromatin remodeling activity. Depletion of ATRX does not alter the levels of cohesin core subunits or positive regulators such as Sororin, nor does it increase Wapl expression, suggesting that ATRX does not affect transcription of cohesin components. Importantly, tethering a Pds5B-binding fragment of ATRX, which lacks the ATPase/helicase domain, rescues cohesion defects in ATRX-depleted cells. This indicates that ATRX's protective role at centromeres is mediated exclusively through its interaction with Pds5B. Furthermore, Wapl depletion bypasses the need for ATRX, emphasizing that ATRX's key function in cohesion maintenance is its ability to antagonize Wapl. These findings provide a detailed mechanistic framework for ATRX's role at centromeres.

ATRX has also been implicated in regulating telomere cohesion, though the data on this remain conflicting. ATRX is frequently lost in cancer cells utilizing the ALT (alternative lengthening of telomeres) pathway to maintain telomeres (Heaphy et al, 2011; Li et al, 2019), and persistent sister-telomere cohesion has been linked to ATRX loss (Ramamoorthy and Smith, 2015). Studies in telomerase-positive glioma cells have shown that ATRX localizes to subtelomeric regions, where its knockdown reduces cohesin occupancy (Eid et al, 2015). ATRX has also been shown to promote sister-telomere cohesion, and its loss in ALT cells compromises telomeric cohesion (Lovejoy et al, 2020). These findings suggest that ATRX coordinates sister chromatid cohesion across chromosomal domains. Our work raises the intriguing possibility that ATRX also interacts with Pds5 at telomeres, and further exploration of this could provide additional insight into its broader contributions to genomic stability.

Beyond its role in cohesion, cohesin is critical for chromatin loop extrusion and the regulation of genome architecture (Hoencamp and Rowland, 2023). Our data show that ATRX binds to Pds5B with comparable efficiency in both G2 phase and mitosis, indicating that the ATRX-Pds5B interaction also exists during interphase. ATRX has been shown to colocalize with cohesin at specific genomic loci, and its loss reduces cohesin occupancy at these sites (Kernohan et al, 2010). These observations raise the possibility that ATRX-bound cohesin may also play a role in modulating DNA looping. Whether ATRX's interaction with Pds5 influences cohesin dynamics in interphase remains an open question, warranting further investigation into its role in chromatin organization.

In conclusion, this study elucidates the molecular mechanism by which ATRX protects centromeric cohesion through competitive inhibition of Wapl. These findings significantly advance our understanding of the pathways that maintain genomic stability during mitosis. Given the frequent mutations in ATRX in cancers and its involvement in regulating sister chromatid cohesion and chromosome segregation, our work may have implications for

understanding tumorigenesis and for developing therapeutic strategies targeting cohesin dynamics. Future studies should investigate whether mutations in ATRX disrupt its interaction with Pds5B and how such changes might contribute to chromosomal instability in cancer.

# Methods

**Reagents and tools table**

| Reagent/Resource | Reference or Source | Identifier or Catalog Number |
| --- | --- | --- |
| **Experimental models** | | |
| HEK-293T cells (*H. sapiens*) | This study | N/A |
| HeLa cells (*H. sapiens*) | Jonathan Higgins lab | N/A |
| **Recombinant DNA** | | |
| Myc-Pds5B | Hongtao Yu Lab | N/A |
| Myc-Pds5A | Hongtao Yu Lab | N/A |
| Myc-SA2 | This study | N/A |
| SMC1A-Myc | This study | N/A |
| SMC3-Myc | This study | N/A |
| RAD21-Myc | This study | N/A |
| HR-SFFV-Tet3G-2A-BFP | Baohui Chen Lab | N/A |
| HR-TRE3G-NLS-dCas9-EGFP | Baohui Chen Lab | N/A |
| sgRNA vector-U6_SP_sgchr15 | Baohui Chen Lab | N/A |
| SFB-ATRX | Junjie Chen Lab | N/A |
| GST-Pds5B (1–300) (WT, F88A, A92P, E94A, I143A, and V138A) | This study | N/A |
| Pds5B-Flag (1–300, 301–1120, and 1121–1447) | This study | N/A |
| Wapl (1–28)-GST (WT, ASE) | This study | N/A |
| MBP-Pds5B (1–300) | This study | N/A |
| pBos-CENP-B (1–163)-GFP | This study | N/A |
| pBos-CB-RAD21-GFP | This study | N/A |
| pBos-CB-ATRX (1394-1443)-GFP (WT, Y1419A) | This study | N/A |
| pBos-ATRX-GFP (WT, R1417A, R1417E, and Y1419A) | This study | N/A |
| pBos-ATRX-GFP (1–845, 846–1540, 1541–2492, 846–1200, 1201–1400, 1401–1540, Δ1394–1443, Δ1444–1494, Δ1495–1540, Δ1413–1443, Δ1394–1421, Δ1422–1443, and 1394–1443) | This study | N/A |
| pBos-Wapl-GFP | This study | N/A |
| GFP-ATRX (1394-1443)–2xStrep (WT, E1413A, E1414A, N1415A, Q1416A, R1417A, S1418A, Y1419A, K1420A, and Q1421A) | This study | N/A |

| Reagent/Resource | Reference or Source | Identifier or Catalog Number |
|---|---|---|
| **Antibodies** | | |
| Rabbit anti-ATRX | Santa Cruz Biotechnology | H300 |
| Rabbit anti-GFP | Invitrogen | A11122 |
| Rabbit anti-GAPDH | Cell Signaling Technology | 14C10 |
| Rabbit anti-RAD21 | Abcam | ab992 |
| Rabbit anti-SMC1A | Bethyl Laboratories | A300-055A |
| Rabbit anti-SMC3 | Bethyl Laboratories | A300-060A |
| Rabbit anti-Pds5B | Bethyl Laboratories | A300-537A, A300-538A |
| Rabbit anti-Sororin | Abcam | ab192237 |
| Rabbit anti-Flag | GenScript | A01868 |
| Mouse anti-α-Tubulin | Sigma | T-6074 |
| Mouse anti-Myc | Millipore | clone 4A6, 05-724 |
| Mouse anti-Flag | Sigma | clone M2, F-3165 |
| Mouse anti-MBP | New England Biolabs | E8032 |
| Mouse anti-SA2 | Santa Cruz Biotechnology | sc-81852 |
| Mouse anti-GFP | Abmart | M20004 |
| Mouse anti-Wapl | MBL | M221-3 |
| Mouse anti-HP1α | Millipore | MAB3446, MAB3584 |
| Mouse anti-H3-pS10 | CST | 6G3 |
| Mouse anti-CENP-A | ENZO | ADI-KAM-CC006-E |
| Human anti-centromere antibody (ACA) | Immunovision | HCT-0100 |
| Guinea pig CENP-C | MBL | PD030 |
| Goat anti-rabbit IgG-HRP | CST | 7074 |
| Horse anti-mouse IgG-HRP | CST | 7076 |
| Donkey anti-rabbit IgG-Alexa Fluor Cy3 | Jackson ImmunoResearch | 711-165-152 |
| Donkey anti-rabbit IgG-Alexa Fluor 488 | Jackson ImmunoResearch | 711-545-152 |
| Goat anti-mouse IgG-Alexa Fluor 488 | Invitrogen | A-32723 |
| Goat anti-mouse IgG-Alexa Fluor 546 | Invitrogen | A-11030 |
| Donkey anti-mouse IgG-Alexa Fluor Cy5 | Jackson ImmunoResearch | 715-175-151 |
| Goat anti-guinea pig IgG-Alexa Fluor 488 | Invitrogen | A-11073 |
| Goat anti-guinea pig IgG-Alexa Fluor 647 | Invitrogen | A-21450 |
| Goat anti-human IgG-Alexa Fluor 647 | Jackson ImmunoResearch | 109-605-003 |

| Reagent/Resource | Reference or Source | Identifier or Catalog Number |
|---|---|---|
| Mouse IgG control | This study | N/A |
| Rabbit IgG control | This study | N/A |
| **Oligonucleotides and other sequence-based reagents** | | |
| siATRX#1 | 5'-GAGGAAACCUUCAAUUGUAUU-3' | RiboBio |
| siATRX#2 | 5'-GUGGGCUGAAGAAUUUAAUdTdT-3' | RiboBio |
| siATRX#3 | 5'-GCAGAGAAAUUCCUAAAGAUU-3' | RiboBio |
| siPds5B | 5'-GAACUUCUACCUUAAGAUUdTdT-3' | RiboBio |
| siWapl | 5'-CGGACUACCCUUAGCACAAdTdT-3' | RiboBio |
| siSororin | 5'-CUCGGAAAGUUUCCUCGCGUAdTdT-3' | RiboBio |
| siSgo1 | 5'-GAGGGGACCCUUUUACAGAdTdT-3' | RiboBio |
| **Chemicals, Enzymes and other reagents** | | |
| STLC | Tocris Bioscience | 2191 |
| MG132 | Selleckchem | S2619 |
| Nocodazole | Selleckchem | S2775 |
| poly-D-lysine | Sigma | 27964-99-4 |
| Thymidine | Calbiochem | 50-89-5 |
| Puromycin | Calbiochem | 58-58-2 |
| Blasticidin | Sigma | 3513-03-9 |
| Doxycycline | Sigma | 24390-14-5 |
| FuGENE 6 | Promega | E2691 |
| Lipofectamine 2000 | Thermo Fisher Scientific | 11668-027 |
| Oligofectamine | Invitrogen | 12252011 |
| Lipofectamine RNAiMAX | Thermo Fisher Scientific | 13778075 |
| **Software** | | |
| GraphPad Prism | https://www.graphpad.com/ | N/A |
| OLYVIA software | https://lifescience.evidentscientific.com | N/A |
| Image J | https://imagej.nih.gov/ij/index.html | N/A |
| **Other** | | |
| AlphaFold3 | https://alphafoldserver.com/welcome | N/A |
| PyMOL | https://pymol.org/ | N/A |
| Uniprot | https://www.uniprot.org/ | N/A |

## Cell lines, DNA constructs, siRNA, transfection, and drug treatments

All cells were cultured in Dulbecco's Modified Eagle Medium (DMEM) supplemented with 10% fetal bovine serum (Gibco) and

1% penicillin-streptomycin, and maintained at 37 °C in a humidified atmosphere containing 5% $CO_2$. HeLa cells stably expressing H2B-GFP, ATRX-GFP (WT or Y1419A), CENP-B-GFP, CENP-B (1394–1443)-ATRX (WT or Y1419A), Sororin-GFP, and Wapl-GFP were cultured in medium supplemented with 2 µg/ml blasticidin (Sigma). HeLa cells stably expressing Myc-Pds5B or Myc-Pds5A were maintained in 0.5 µg/ml puromycin (Calbiochem).

The constructs for Myc-Pds5B and Myc-Pds5A were provided by Hongtao Yu (Westlake University). The SFB-ATRX plasmid was provided by Dr. Junjie Chen (The University of Texas MD Anderson Cancer Center). Constructs for HR-SFFV-Tet3G-2A-BFP, HR-TRE3G-NLS-dCas9-EGFP, and U6_SP_sgchr15 were provided by Dr. Baohui Chen (Zhejiang University).

The cDNAs encoding SA2, SMC1A, SMC3, and RAD21 were amplified by PCR and subcloned into expression vectors containing a Myc epitope tag. Plasmids encoding Pds5B-Flag variants were generated by subcloning PCR-amplified fragments corresponding to Pds5B residues 1–300, 301–1120, and 1121–1447 into the pEF6-Flag vector. The pBos-CENP-B-GFP plasmid was constructed by replacing the H2B coding sequence in pBos-H2B-GFP (BD PharMingen) with the KpnI/BamHI-digested PCR fragments encoding the centromere-targeting domain of CENP-B (residues 1–163). The pBos-CB-RAD21-GFP construct was generated by subcloning RAD21 PCR fragments into pBos-CENP-B-GFP, and pBos-CB-ATRX (1394–1443)-GFP constructs were created by subcloning ATRX fragments into the same vector. The pBos-ATRX-GFP plasmid series was constructed by replacing the H2B coding sequence in pBos-H2B-GFP with PCR-amplified full-length ATRX or its variant fragments (1–845, 846–1540, 1541–2492, 846–1200, 1201–1400, 1401–1540, Δ1394–1443, Δ1444–1494, Δ1495–1540, Δ1413–1443, Δ1394–1421, Δ1422–1443, and 1394–1443). The pBos-Wapl-GFP construct was generated by replacing the H2B fragment with the PCR-amplified Wapl coding sequence. The plasmid encoding GST-fused Pds5B (1–300) was generated by subcloning PCR fragments into pGEX-4T1 (GE Healthcare). For the Wapl (1–28)-GST construct, PCR fragments were cloned into pETGEX-CT. The MBP-Pds5B (1–300) construct was generated by cloning PCR products into pMAL-C2E (New England Biolabs). To generate the GFP-ATRX (1394–1443)-2xStrep plasmid, the TAZ sequence in GFP-TAZ-2xStrep was replaced with the PCR-amplified ATRX (1394–1443) fragment.

All point mutations were introduced using either the QuikChange II XL Site-Directed Mutagenesis Kit (Agilent Technologies) or the MultiF Seamless Assembly Mix Kit (ABclonal Biotechnology). All constructs were verified by DNA sequencing to confirm the presence of desired mutations and the absence of unwanted sequence alterations.

siRNA duplexes targeting ATRX, Wapl, Pds5B, Sororin, and Sgo1 were purchased from Integrated DNA Technologies (IDT) or RiboBio. siRNA transfections were carried out twice at a 24 h interval using Oligofectamine (Invitrogen) or Lipofectamine RNAiMAX (Thermo Fisher Scientific). Plasmid transfections were performed using FuGENE 6 (Promega) or Lipofectamine 2000 (Thermo Fisher Scientific).

Cell cycle synchronization was achieved as follows: cells were synchronized in S or G2 phase via double thymidine block (2 mM; Calbiochem); in prometaphase using STLC (5 µM; Tocris Bioscience) or nocodazole (100 ng/ml; Selleckchem); in metaphase by treatment with MG132 (10 µM; Selleckchem); and in G1 phase by release from nocodazole-induced arrest. Mitotic cells were collected by mechanical detachment ("shake-off").

## Antibodies

The following rabbit polyclonal antibodies were used: ATRX (H300, Santa Cruz Biotechnology), GFP (A11122, Invitrogen), GAPDH (14C10, Cell Signaling Technology/CST), RAD21 (ab992, Abcam), SMC1A (A300-055A, Bethyl Laboratories), SMC3 (A300-060A, Bethyl Laboratories), Pds5B (A300-537A and A300-538A, Bethyl Laboratories), Sororin (ab192237, Abcam), and Flag tag (A01868, GenScript). Mouse monoclonal antibodies included: α-Tubulin (T-6074, Sigma), Myc tag (clone 4A6, 05-724, Millipore), Flag tag (clone M2, F-3165, Sigma), MBP (E8032, New England Biolabs), SA2 (sc-81852, Santa Cruz Biotechnology), GFP (M20004, Abmart), Wapl (M221-3, MBL), HP1α (MAB3446 for immunoblotting and MAB3584 for immunostaining, Millipore), histone H3 phosphorylated at serine 10 (H3-pS10; 6G3, CST), and CENP-A (ADI-KAM-CC006-E, ENZO). The anti-centromere antibody (ACA) was obtained from ImmunoVision (HCT-0100). Guinea pig polyclonal antibodies against CENP-C were purchased from MBL (PD030).

Secondary antibodies for immunoblotting included goat anti-rabbit IgG-HRP (7074, CST) and horse anti-mouse IgG-HRP (7076, CST). Secondary antibodies for immunofluorescence staining included: donkey anti-rabbit IgG conjugated to Alexa Fluor 488 or Cy3 (711-545-152 or 711-165-152, Jackson ImmunoResearch); anti-mouse IgG conjugated to Alexa Fluor 488, 546 (A-32723 or A-11030, Invitrogen) or Cy5 (715-175-151, Jackson ImmunoResearch); goat anti-guinea pig IgG conjugated to Alexa Fluor 488 or 647 (A-11073 or A-21450, Invitrogen); and anti-human IgG conjugated to Alexa Fluor 647 (109-605-003, Jackson ImmunoResearch).

## Fluorescence microscopy and quantification

HeLa cells were fixed in 2% paraformaldehyde (PFA) in phosphate-buffered saline (PBS) for 10 min at room temperature, followed by permeabilization with 0.5% Triton X-100 in PBS for 5 min. Cells were subsequently blocked with 3% bovine serum albumin (BSA) in PBS and incubated with primary antibodies for 1–2 h, followed by secondary antibody incubation for 1 h, both at room temperature in 3% BSA/PBS. Nuclear DNA was counterstained with DAPI for 5 min. For chromosome spread preparation, HeLa cells were treated with either 10 µM MG132 for 2–8 h or 100 ng/ml nocodazole for 3 h. Mitotic cells were collected by selective detachment and incubated in hypotonic buffer (75 mM KCl or 0.25xPBS) for 10–15 min at room temperature. Cells were then cytocentrifuged onto glass coverslips using a Cytospin at 1500 rpm for 5 min. Following attachment, chromosome spreads were fixed in 2% PFA in PBS for 15–20 min, permeabilized with 0.5% Triton X-100 in PBS for 10 min, blocked with 3% BSA in PBS, and incubated with primary antibodies for 2 h at room temperature. For ATRX immunofluorescence detection, cells were fixed in 3:1 ethanol/methanol mixture for 30 min, followed by permeabilization with 0.2% Triton X-100 in PBS for 15 min. Fluorescence microscopy was performed at room temperature using a Nikon ECLIPSE Ni microscope equipped with a Plan Apo Fluor 60x oil immersion objective (NA 1.4) and a Clara CCD camera (Andor Technology).

Inter-KT distances were measured using NIS-Elements BR software by quantifying the distance between CENP-C, ACA, or CENP-A immunofluorescence signals on at least 25 kinetochore pairs per cell, across a minimum of 20 cells per experiment. Sister chromatids that were visibly separated were excluded from the analysis. Fluorescence intensity quantification was conducted using ImageJ (NIH) on images acquired under identical illumination and exposure conditions. Circular regions encompassing paired centromeres were selected, and the average pixel intensity for ACA, ATRX, or Sororin-GFP signals was measured. After background subtraction, the ratios of centromeric ATRX/ACA and Sororin-GFP/ACA fluorescence intensities were calculated for each centromere.

## Three-dimensional structured illumination microscopy (3D-SIM)

Fluorophores were excited using laser lines at 405 nm, 488 nm, and 642 nm. 3D-SIM images were acquired with a DeltaVision OMX system (GE Healthcare) equipped with a 60x oil immersion objective (NA 1.42). Z-stacks were collected at 125 nm intervals to cover the full cellular thickness. Raw SIM data were reconstructed using SoftWoRx 7.0 software (GE Healthcare).

## Time-lapse live cell imaging

Time-lapse imaging was performed using an Olympus IXplore SpinSR spinning disk confocal system equipped with a 60x oil immersion objective and controlled via OLYVIA software. HeLa cells stably expressing H2B-GFP were seeded onto poly-D-lysine-coated four-chamber glass-bottom 35-mm dishes (Cellvis) and maintained in a climate-controlled, humidified chamber at 37 °C with 5% $CO_2$. Images were acquired at 2.5 min intervals and subsequently processed and assembled using Adobe Photoshop and Illustrator.

## Protein expression and purification

Plasmids encoding GST-, MBP-, or Strep-fused proteins were transformed into *E. coli* BL21 (DE3) competent cells. Transformed cells were cultured in LB medium supplemented with the appropriate antibiotic at 37 °C until reaching an optical density at 600 nm ($OD_{600}$) of 0.6–0.8. Protein expression was induced by adding 0.4 mM isopropyl β-D-1-thiogalactopyranoside (IPTG) and incubation at 16 °C for 16 h.

For GST fusion proteins, cells were lysed by sonication in Buffer A (20 mM Tris-HCl, pH 8.0, 100 mM NaCl, 1 mM EDTA, 1% Triton X-100). For MBP fusion proteins, lysis was performed in Buffer B (50 mM Tris-HCl, pH 7.5, 300 mM NaCl, 1 mM EDTA, 0.5% Triton X-100). Lysates were clarified by centrifugation and incubated with Glutathione Sepharose 4B beads (GE Healthcare) for GST fusion proteins or Amylose Resin (New England Biolabs) for MBP fusion proteins. After washing with the respective lysis buffer, proteins were eluted using 100 mM reduced glutathione (for GST fusions) or 10 mM maltose (for MBP fusions).

For Strep-fused protein purification, cells were resuspended in lysis buffer (25 mM Tris-HCl, pH 7.5, 500 mM NaCl, 1 mM DTT, 1% Triton X-100, 1 mM EDTA, and 1 mM phenylmethylsulfonyl fluoride (PMSF)). Following sonication and a freeze-thaw cycle, the lysate was clarified by centrifugation, and the supernatant was applied to a Strep-Tactin Superflow affinity column (IBA Life-sciences). The resin was washed four times with lysis buffer, and bound proteins were eluted with lysis buffer supplemented with 10 mM desthiobiotin.

## Pull-down, immunoprecipitation, and immunoblotting

For GST- or Strep-fused protein pull-down assays with cell lysates, HeLa cells were lysed in P100 buffer (50 mM Tris-HCl, pH 7.5, 100 mM NaCl, 1% Triton X-100, 10 mM $MgCl_2$, 5 mM EDTA) supplemented with 1 mM dithiothreitol (DTT), protease inhibitor cocktail (P8340, Sigma), 1 mM PMSF, 0.1 μM okadaic acid (Calbiochem), 10 mM sodium fluoride, 20 mM β-glycerophosphate, and Benzonase (GenScript). Insoluble material was removed by high-speed centrifugation, and lysates were precleared with either glutathione Sepharose 4B beads (GE Healthcare) or StrepTactin 4FF beads (Smart Life Sciences). Precleared lysates were incubated with bead-immobilized GST- or Strep-fused proteins for 4 h at 4 °C. Beads were washed three times with lysis buffer, boiled in SDS sample buffer, and subjected to immunoblotting.

For the competitive binding assays, cell lysates or purified MBP-Pds5B were mixed with recombinant GFP-2×Strep or GFP-ATRX (1394–1443)-2xStrep (WT or Y1419A) and subjected to pull-down using Wapl (1–28)-GST.

For co-immunoprecipitation, cells were lysed in P50 buffer (50 mM Tris-HCl, pH 7.5, 50 mM NaCl, 1% Triton X-100, 10 mM $MgCl_2$, 5 mM EDTA) supplemented with the same additives as above. Insoluble material was removed by high-speed centrifugation, and lysates were precleared with rProtein A/G beads (Smart Life Sciences, Cat. No. SA032100). Precleared lysates were incubated for 4 h at 4 °C with Anti-GFP Affinity beads (Smart Life Sciences, SA070005), Anti-Flag M2 Affinity Gel (Sigma, A2220), Anti-Myc Affinity Gel (GNI, GNI4510-MC), or control IgG. For endogenous ATRX co-immunoprecipitation, precleared lysates were incubated with 2 μg/ml ATRX antibodies or rabbit IgG for 3 h at 4 °C, followed by incubation with rProtein A/G beads for 1 h at 4 °C. After washing with P50 buffer, bound proteins were eluted by boiling in SDS sample buffer for immunoblotting.

## Mass spectrometry

For the data presented in Dataset EV1 and Dataset EV2, co-immunoprecipitated IgG and Myc-Pds5B-binding proteins were separated by SDS-PAGE. Protein bands were excised, cut into small pieces, and de-stained with buffer (25 mM $NH_4HCO_3$, 25% methanol, pH 8.0). Proteins were reduced with 10 mM DTT for 60 min at 56 °C and alkylated with 55 mM iodoacetamide for 45 min. Gel pieces were washed twice with digestion buffer (50 mM $NH_4HCO_3$, pH 8.0), dehydrated with acetonitrile, and dried in a SpeedVac. After rehydration with trypsin solution (10 ng/μl sequencing-grade modified trypsin in 50 mM $NH_4HCO_3$, pH 8.0), samples were incubated overnight at 37 °C.

Digested peptides were extracted sequentially with Elution Buffer I (50% acetonitrile, 5% formic acid) and Elution Buffer II (75% acetonitrile, 0.1% formic acid). Gel pieces were further dehydrated twice with acetonitrile, and supernatants were pooled.

The peptide solution was dried in a SpeedVac, resuspended in 5% formic acid, desalted using StageTips, and loaded onto an analytical column (75 × 150 mm, 1.9 μm C18, 1 μm tip) with an Easy-nLC 1200 system. Peptides were separated using a 60 min gradient at a flow rate of 300 nL/min as follows: 3–6% B for 2 min, 6–26% B for 38 min, 26–34% B for 12 min, 34–90% B for 3 min, and 90–100% B for 3 min.

The Orbitrap Exploris 480 mass spectrometer was operated in data-dependent mode with one full MS scan at a resolution of 60,000 ($m/z$ 200), followed by twenty HCD MS/MS scans at a resolution of 15,000 (normalized collision energy, NCE = 30; isolation width of 1.6 $m/z$). Precursors with charge states of +2 to +5 were selected; isotope exclusion was applied, and dynamic exclusion was set to 30 s. Mass spectrometry data were analyzed using MaxQuant.

### Labeling of endogenous genomic loci using the CRISPR/dCas9 system

The catalytically inactive Cas9 (dCas9) was used for CRISPR imaging. Cells were transfected with the following plasmids: pHR-SFFV-Tet3G-2A-BFP, pHR-TRE3G-NLS-dCas9-EGFP, and the sgRNA expression vector U6_SP_sgchr15. Twenty-four hours post-transfection, the medium was replaced with fresh culture medium containing 1 μg/ml doxycycline, and cells were incubated for an additional 24 h. Following release from a double thymidine block, G2-phase cells were harvested 8 h later and subjected to immunofluorescence staining for GFP and DAPI. The distance between sister loci in doublets was quantified using NIS-Elements BR software (Nikon), based on the GFP immunofluorescence signal. The target sequence for the sgchr15 guide RNA was: 5′-GACGCAGATGAGGGACGGG-3′.

### Structure prediction

The structure of the Pds5B N-terminal fragment (residues 1–300) in complex with the ATRX-derived peptide RSYKQ was predicted using AlphaFold3. Structural figures were generated using PyMOL.

### Sequence alignment

Multiple sequence alignment of ATRX, Wapl, Sororin, Haspin, and PD-L1 was performed using CLUSTALW (https://www.genome.jp/tools-bin/clustalw), and the results were visualized using ESPript 3.0 (https://espript.ibcp.fr/ESPript/ESPript/).

### Statistical analysis

Statistical analyses were conducted using a two-tailed unpaired Student's t-test in GraphPad Prism 10. A *P*-value of <0.05 was considered statistically significant. *P* values are denoted as follows: *$p < 0.05$, **$p < 0.01$, ***$p < 0.001$, ****$p < 0.0001$.

## Data availability

This study includes no data deposited in external repositories.

The source data of this paper are collected in the following database record: biostudies:S-SCDT-10_1038-S44318-025-00465-6.

## Peer review information

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

## Acknowledgements

We thank Drs. Bohui Chen, Junjie Chen, and Hongtao Yu for kindly providing plasmids, and our lab members for helpful discussions. This work was supported by the National Outstanding Youth Science Fund Project of National Natural Science Foundation of China (32025011 to FW), the National Key R&D Program of China (2022YFA1303102 and 2022YFA1105203 to FW), the National Natural Science Foundation of China (32270772 to HY, 32061160470 to FW, 32100583 to QC, and 324B2019 to LZ), the Natural Science Foundation of Zhejiang Province (LZ24C070001 to FW), and the Central Guidance for Local Scientific and Technological Development Funding Project (2025ZY01106 to FW). We also acknowledge the Life Sciences Institute Core Facility and Dr. Cheng Ma at the Core Facility of Zhejiang University School of Medicine.

## Author contributions

**Lei Zhao**: Data curation; Software; Funding acquisition; Validation; Investigation; Visualization; Methodology; Writing—review and editing. **Xueying Yuan**: Data curation; Software; Validation; Investigation; Methodology; Writing—review and editing. **Qinfu Chen**: Funding acquisition; Investigation; Methodology; Writing—review and editing. **Haiyan Yan**: Conceptualization; Resources; Data curation; Software; Formal analysis; Supervision; Funding acquisition; Validation; Investigation; Visualization; Methodology; Writing—original draft; Project administration; Writing—review and editing. **Fangwei Wang**: Conceptualization; Resources; Data curation; Software; Formal analysis; Supervision; Funding acquisition; Validation; Investigation; Visualization; Methodology; Writing—original draft; Project administration; Writing—review and editing.

Source data underlying figure panels in this paper may have individual authorship assigned. Where available, figure panel/source data authorship is listed in the following database record: biostudies:S-SCDT-10_1038-S44318-025-00465-6.

## Disclosure and competing interests statement

The authors declare no competing interests.

# Expanded View Figures

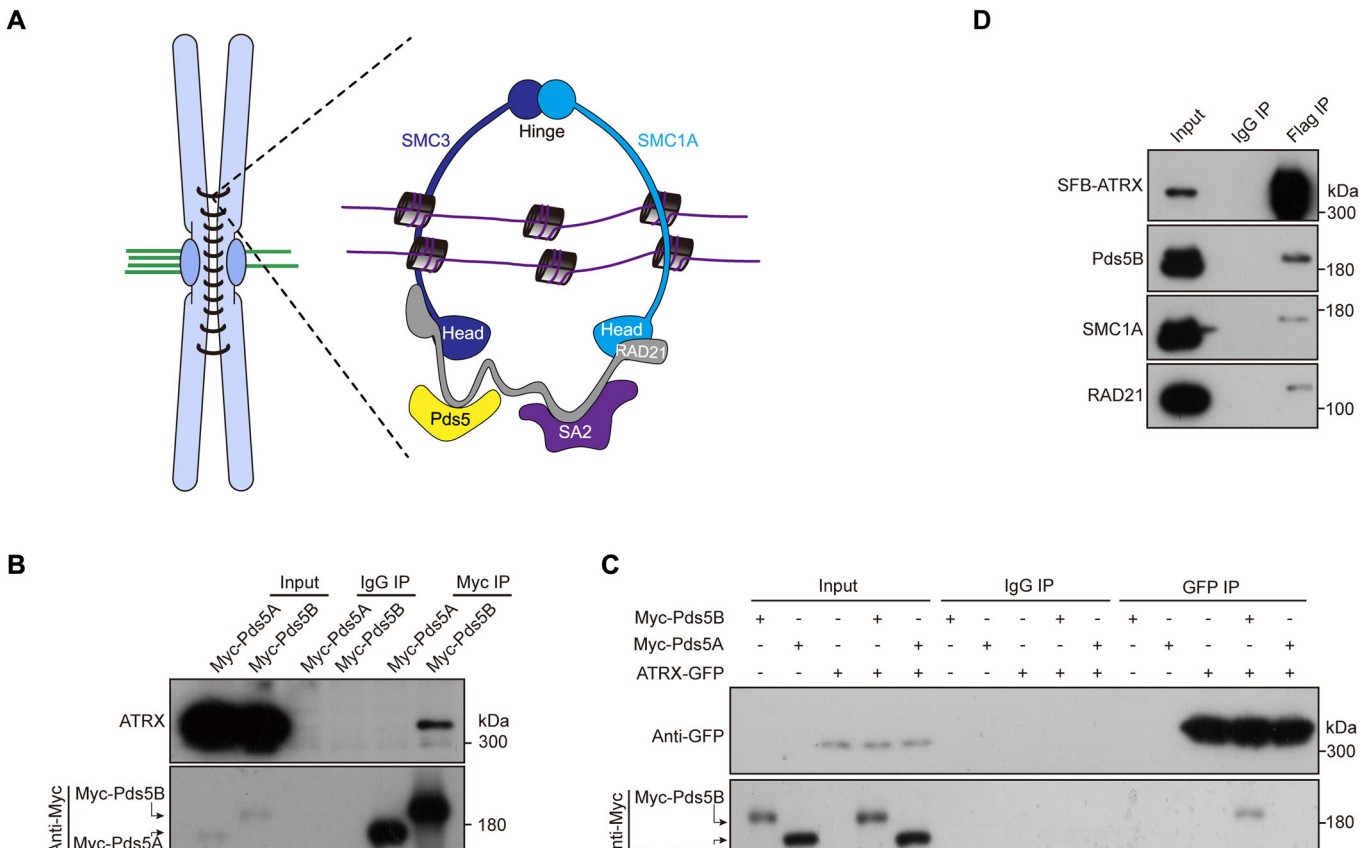

**Figure EV1.   A proteomic screen identifies ATRX as a Pds5B-associated factor.**

(A) Schematic diagram illustrating sister chromatid cohesion mediated by the ring-shaped cohesin complex. (B) Nocodazole-arrested mitotic HeLa cells stably expressing Myc-Pds5A or Myc-Pds5B were immunoprecipitated using anti-Myc beads or control IgG, followed by immunoblotting for ATRX and the Myc tag. (C) HEK-293T cells co-expressing ATRX-GFP with Myc-Pds5A or Myc-Pds5B were immunoprecipitated using anti-GFP beads or control IgG, followed by immunoblotting for GFP and the Myc tag. (D) HEK-293T cells expressing SFB-ATRX were immunoprecipitated using anti-Flag beads or control IgG, followed by immunoblotting for the Flag tag, Pds5B, SMC1A, and RAD21.

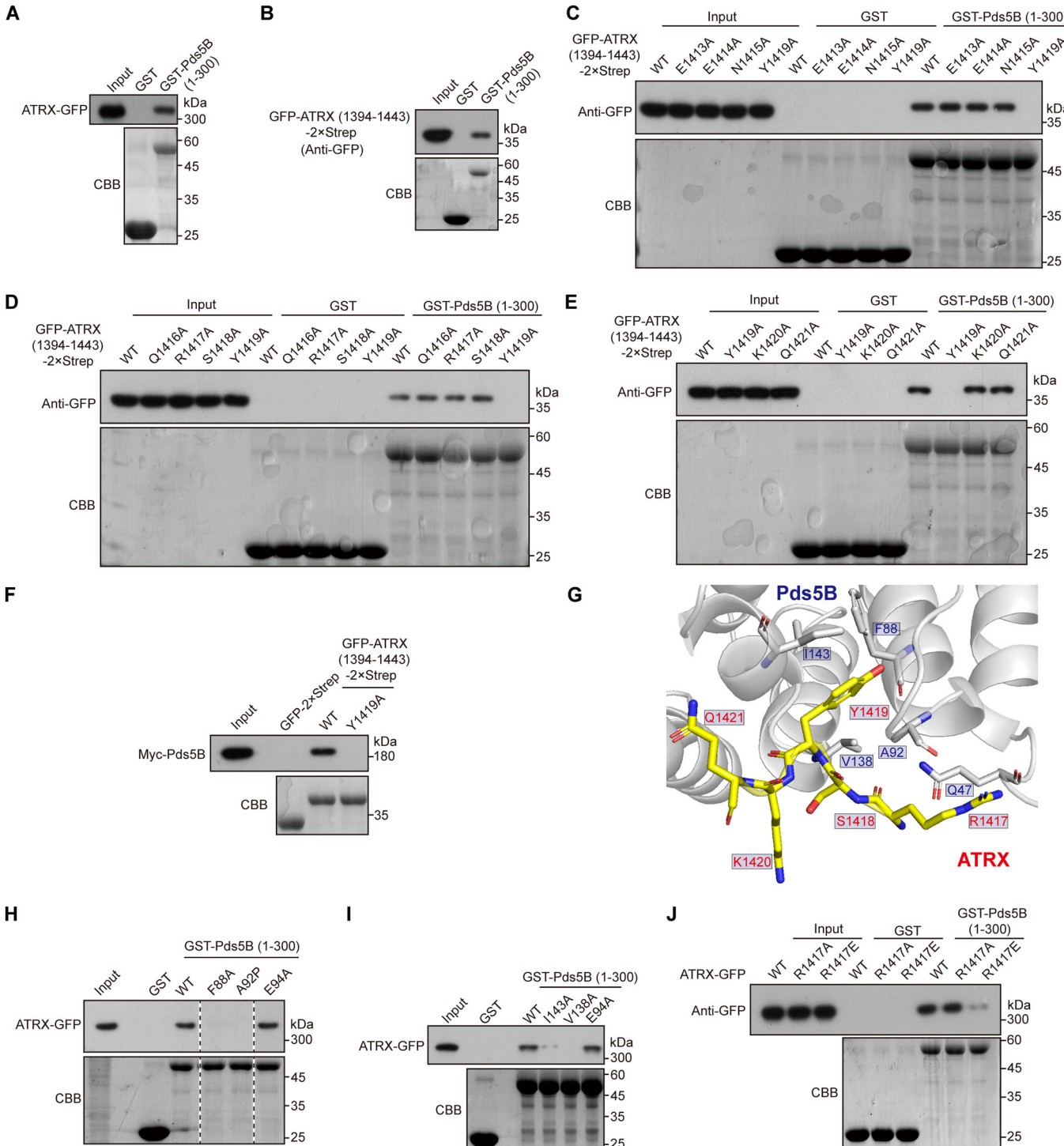

**Figure EV2. Mapping the molecular interface between ATRX and Pds5B.**

(A) HEK-293T cells expressing ATRX-GFP were subjected to pull-down using GST or GST-Pds5B (1–300), followed by immunoblotting for GFP and CBB staining. (B–E) Recombinant GFP-ATRX (1394–1443)-2xStrep (wild-type or mutants) proteins were subjected to pull-down using GST or GST-Pds5B (1–300), followed by immunoblotting for GFP and CBB staining. (F) HEK-293T cells expressing Myc-Pds5B were subjected to pull-down using GFP-2xStrep or GFP-ATRX (1394–1443)-2xStrep (WT or Y1419A), followed by immunoblotting for the Myc tag, and CBB staining. (G) Cartoon presentation of the structure for ATRX (RSYKQ) binding to Pds5B (1–300). ATRX and Pds5B residues are shown in yellow and gray, respectively. The binding details of ATRX residues R1417, S1418, Y1419, K1420, and Q1421 with Pds5B residues Q47, F88, A92, V138, and I143 are depicted. (H, I) HEK-293T cells expressing ATRX-GFP were subjected to pull-down using GST or GST-Pds5B (1–300) (WT or mutants), followed by immunoblotting for GFP and CBB staining. (J) HEK-293T cells expressing ATRX-GFP (WT, R1417A or R1417E) were subjected to pull-down using GST or GST-Pds5B (1–300), followed by immunoblotting for GFP and CBB staining.

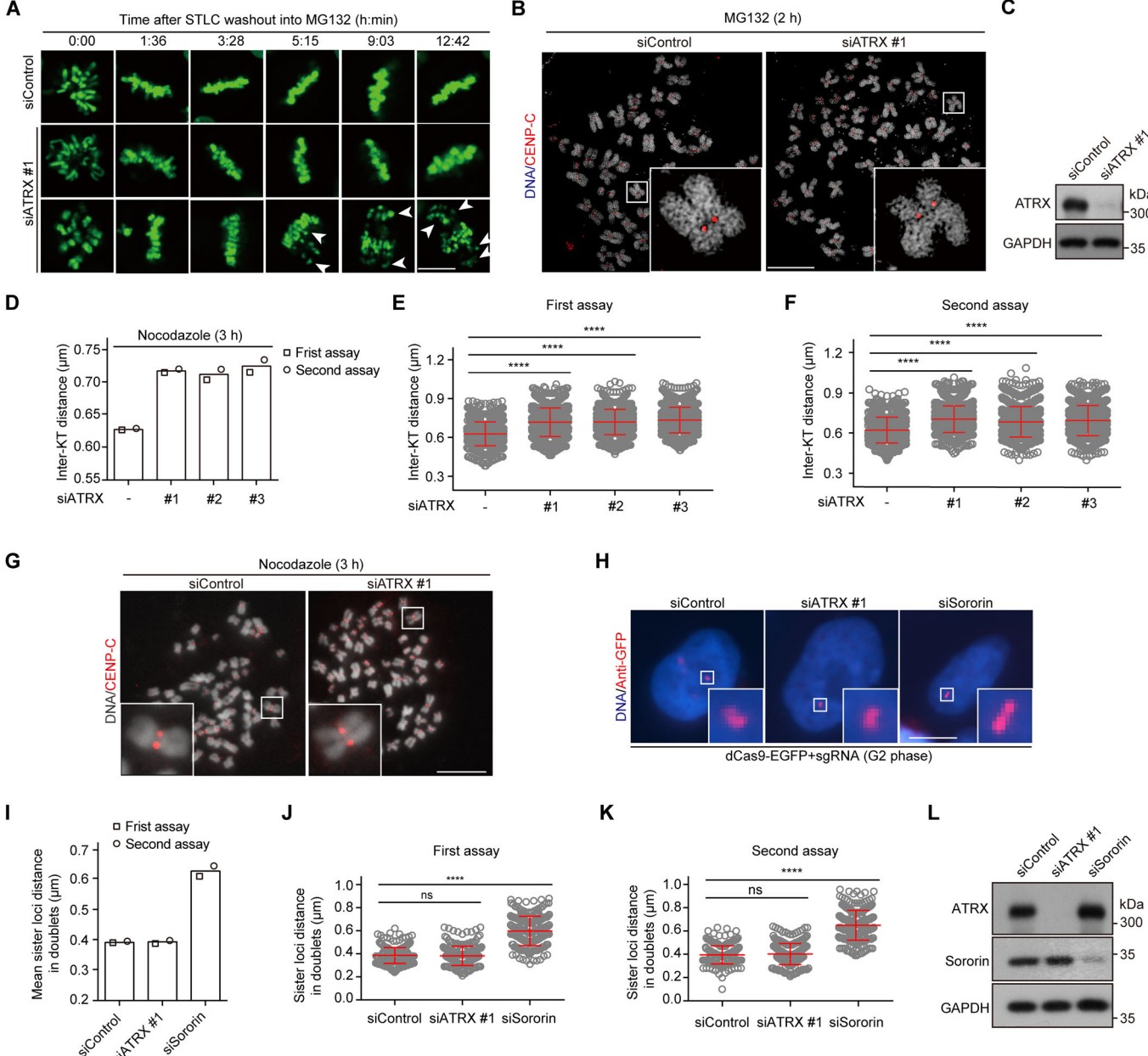

**Figure EV3.  ATRX promotes centromeric cohesion and prevents chromosome missegregation.**

(A) HeLa cells stably expressing H2B-GFP were transfected with control or ATRX siRNA, followed by live imaging of mitosis progression. Selected movie frames are shown. Time is indicated in hours: minutes. Arrows point to scattering chromosomes. Related to Fig. 3D. (B, C) HeLa cells were transfected with control or ATRX siRNA. Forty-eight hours post-transfection, cells were treated with MG132 for 2 h, and mitotic chromosome spreads were stained for CENP-C and DAPI. Representative super-resolution images are shown (B). Cell lysates were analyzed by immunoblotting for ATRX and GAPDH (C). (D–G) HeLa cells transfected as described above were treated with nocodazole for 3 h, and mitotic chromosome spreads were stained for CENP-C and DAPI. The inter-KT distance was measured on more than 2000 chromosomes from 95 cells per condition across two independent experiments, and the means and ranges are plotted (D). Inter-KT distances from two individual experiments are shown. $p$ values from left to right: ****$p < 1.00E-15$, ****$p < 1.00E-15$, ****$p < 1.00E-15$ (E). ****$p < 1.00E-15$, ****$p < 1.00E-15$, ****$p < 1.00E-15$ (F). Representative images are shown (G). (H–L) HeLa cells were transfected with the indicated siRNAs, co-transfected with NLS-dCas9-EGFP-expressing vector and the sgRNA-expressing vector, and synchronized using double thymidine treatment. Eight hours after release from thymidine, cells were stained for GFP and DAPI. Representative images are shown (H). The sister loci distance was measured in more than 450 cells per condition across two independent experiments, and the means and ranges are plotted (I). Sister loci distances from two individual experiments are shown. $p$ values from left to right: ns $p = 6.12E-01$, ****$p < 1.00E-15$ (J). ns $p = 3.95E-01$, ****$p < 1.00E-15$ (K). Cell lysates were analyzed by immunoblotting for ATRX, Sororin, and GAPDH (L). Data information: Statistics were performed using unpaired Student's t-test (E, F, J, K). Means and SDs are shown (E, F, J, K). ns, no significance. Scale bars, 10 μm (A, B, G, H).

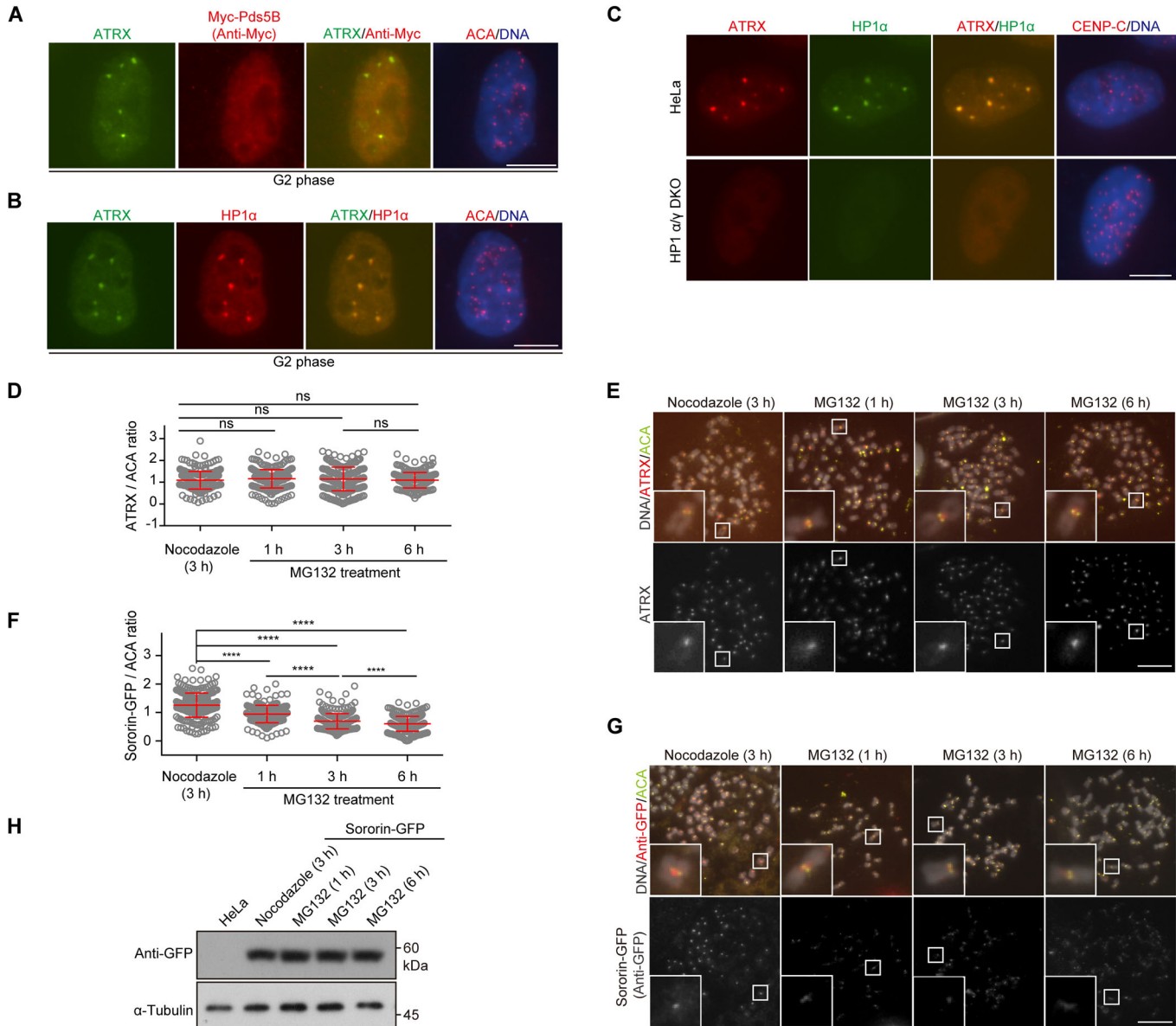

**Figure EV4. ATRX is enriched at mitotic centromeres in an HP1-dependent manner.**

(A) HeLa cells stably expressing Myc-Pds5B were synchronized using double thymidine treatment. Eight hours after release from thymidine, cells were stained for ATRX, the Myc tag, ACA, and DAPI. Representative images are shown. (B) HeLa cells were synchronized in G2 phase as described above and stained for ATRX, HP1α, ACA, and DAPI. Representative images are shown. (C) HeLa and HP1α/γ DKO cells were stained for ATRX, HP1α, CENP-C, and DAPI. Representative images are shown. (D, E) HeLa cells were treated with nocodazole for 3 h or with MG132 for 1, 3, and 6 h, and mitotic chromosome spreads were stained for ATRX, ACA, and DAPI. The fluorescence intensity ratio of ATRX to ACA was quantified on more than 230 chromosomes from 15–20 cells. *p* values from left to right: ns *p* = 6.59E−02, ns *p* = 1.91E−01, ns *p* = 9.35E−01, ns *p* = 2.07E−01 (D). Representative images are shown (E). (F–H) HeLa cells stably expressing Sororin-GFP were treated with nocodazole for 3 h or with MG132 for 1, 3, and 6 h, and mitotic chromosome spreads were stained for GFP, ACA, and DAPI. The fluorescence intensity ratio of Sororin-GFP to ACA was quantified on more than 195 chromosomes from 15 to 20 cells. *p* values from left to right: ****$p$ < 1E−15, ****$p$ < 1E−15, ****$p$ < 1E−15, ****$p$ < 1E−15 (F). Representative images are shown (G). Cell lysates were analyzed by immunoblotting for GFP and α-Tubulin (H). Data information: Statistics were performed using unpaired Student's t-test (D, F). Means and SDs are shown (D, F). N.S., no significance. Scale bars, 10 μm (A, B, C, E, G).

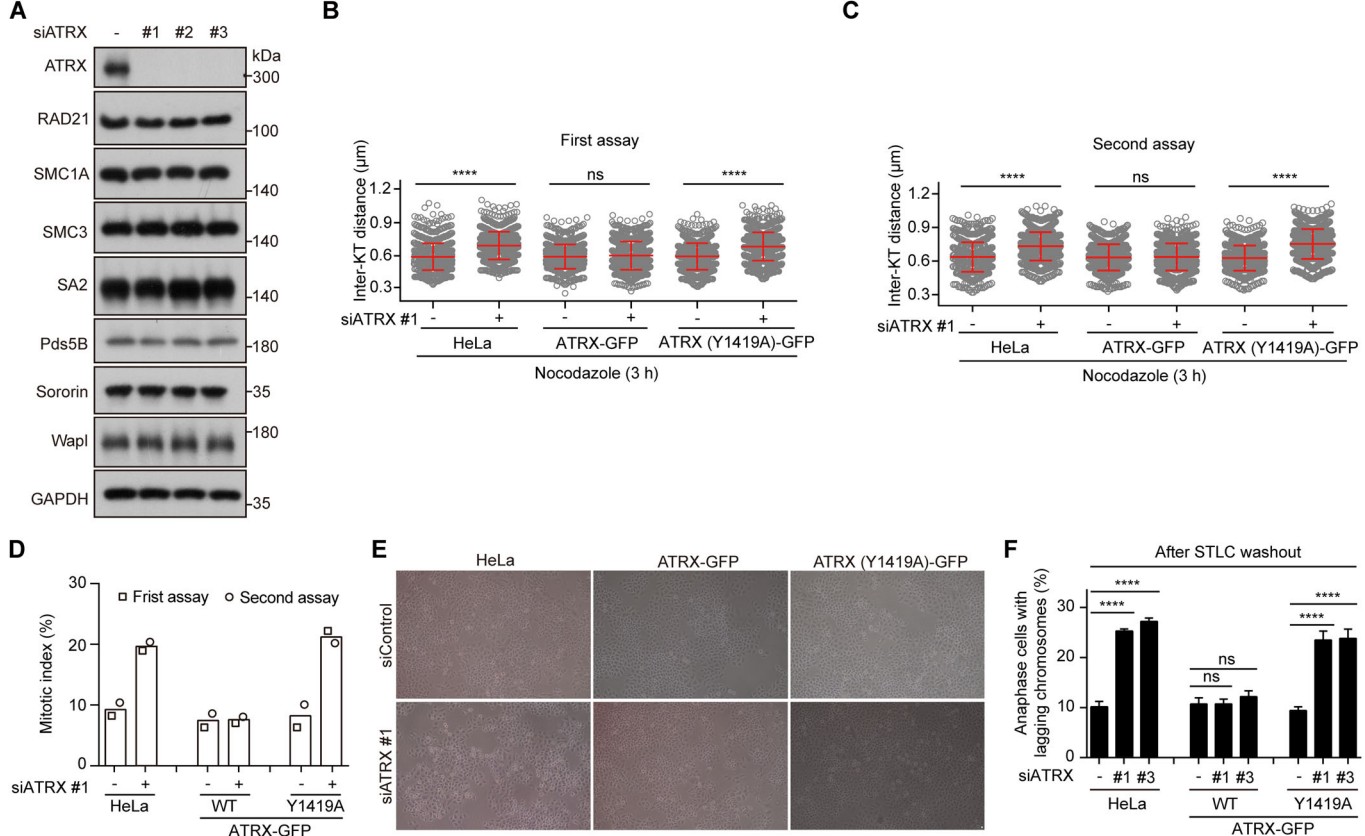

**Figure EV5. ATRX maintains centromeric cohesion through interaction with Pds5B.**

(A) HeLa cells were transfected with control or ATRX siRNAs. Forty-eight hours post-transfection, cells were analyzed by immunoblotting for ATRX, RAD21, SMC1A, SMC3, SA2, Pds5B, Sororin, Wapl, and GAPDH. (B, C) HeLa and HeLa cells stably expressing ATRX-GFP (WT or Y1419A) were transfected as described above, treated with nocodazole for 3 h, and mitotic chromosome spreads were stained for GFP, ACA, and DAPI. The inter-KT distance was measured on more than 1750 chromosomes from 80 cells per condition across two independent experiments. *p* values from left to right: ****$p < 1.00E{-}15$. ns $p = 1.01E{-}01$, ****$p < 1.00E{-}15$ (B). ****$p < 1.00E{-}15$ (J). ns $p = 5.10E{-}01$, ****$p < 1.00E{-}15$ (C). Related to Fig. 5D. (D, E) HeLa and HeLa cells stably expressing ATRX-GFP (WT or Y1419A) were transfected with control or ATRX siRNA. Forty-eight hours post-transfection, cells were subjected to mitotic index analysis. The mitotic index was determined from more than 940 cells per condition across two independent experiments (D). Representative images of cells are shown, with round cells classified as mitotic (E). (F) HeLa and HeLa cells stably expressing ATRX-GFP (WT or Y1419A) transfected as described above were treated with STLC for 5 h. Two hours after STLC release, cells were fixed and stained with DAPI. The percentage of anaphase cells with lagging chromosomes was quantified from more than 560 cells per condition across three independent experiments. *p* values from left to right: ****$p = 5.30E{-}05$. ****$p = 5.00E{-}05$, ns $p = 9.92E{-}01$, ns $p = 2.09E{-}01$. ****$p = 2.44E{-}04$. ****$p = 2.29E{-}04$. Data information: Statistics were performed using unpaired Student's t-test (B, C, F). Means and SDs are shown (B, C, F). ns, no significance. Scale bars, 10 μm (E).

