## [Peer Review File · The EMBO Journal]

A chromatin-remodeling-independent role for ATRX in protecting centromeric cohesion

Lei Zhao, Xueying Yuan, Qinfu Chen, Haiyan Yan, and Fangwei Wang

Corresponding author(s): Fangwei Wang (fwwang@zju.edu.cn)

Review Timeline:

Submission Date:	13th Jan 25
Editorial Decision:	20th Feb 25
Revision Received:	14th Apr 25
Accepted:	13th May 25

Editor: Hartmut Vodermaier

Transaction Report:

Prof. Fangwei Wang
Zhejiang University
Life Sciences Institute
866 Yuhangtang Rd
Nano Building Rm 577
Hangzhou, Zhejiang 310058
China

20th Feb 2025

Re: EMBOJ-2025-120195-T
A chromatin remodeling-independent role for ATRX in protecting centromeric cohesion

Dear Fangwei,

Thank you for submitting your study on an ATRX role in centromeric cohesion protection to The EMBO Journal. It has now been seen by three expert referees, whose reports are copied below for your information. As you will see, the reviewers consider the implication of an additional Pds5B interactor and Wapl antagonist potentially interesting, but also ask questions about its overall importance and relative significance compared to other protection mechanisms. Furthermore, they all raise a variety of well-taken specific issues, but also provide constructive suggestions for how these may be addressed. In this light, I would be happy to give you an opportunity to respond to the referees' concerns via a revised version of the manuscript.

Since we only allow for a single round of major revision, the suitability of this work for EMBO Journal publication will ultimately depend on whether the key concerns of the reviewers could be satisfactorily clarified at the stage of resubmission. In this light, I would very much encourage you to contact me with a revision plan and preliminary point-by-point response already during the early stages of your revision work, so that we could discuss if and how the main points could be resolved, or whether a less completely revised manuscript might alternatively become suitable for publication in one of our sister journals like EMBO Reports or Life Science Alliance. Of course, we would also be open to extension of the default three-months revision period if needed; our 'scooping protection' (meaning that competing work appearing elsewhere in the meantime will not affect our considerations of your study) would of course remain valid also throughout such an extension.

Detailed information on preparing, formatting and uploading a revised manuscript can be found below and in our Guide to Authors. Thank you again for the opportunity to consider this work for The EMBO Journal, and I look forward to hearing from you in due time.

With kind regards,

Hartmut

- size of the scale bars that are mandatory for all micrograph panels
- the statistical test used to generate error bars and P-values
- the type error bars (e.g., S.E.M., S.D.)
- the number (n) and nature (biological or technical replicate) of independent experiments underlying each data point
- Figures may not include error bars for experiments with $n < 3$; scatter plots showing individual data points should be used

instead.

9) To facilitate reproducibility and cross-laboratory adoption of methodologies, please structure the Materials & Methods section as outlined in our guide to authors, including a completed Reagents and Tools Table that can be downloaded from our author guidelines as well (<https://www.embopress.org/page/journal/14602075/authorguide#structuredmethods>).

10) Digital image enhancement is acceptable practice, as long as it accurately represents the original data and conforms to community standards. If a figure has been subjected to significant electronic manipulation, this must be clearly noted in the figure legend and/or the 'Materials and Methods' section. The editors reserve the right to request original versions of figures and the original images that were used to assemble the figure. Finally, we generally encourage uploading of numerical as well as gel/blot image source data; for details see: embopress.org/page/journal/14602075/authorguide#sourcedata

At EMBO Press, we ask authors to provide source data for the main manuscript figures. Our source data coordinator will contact you to discuss which figure panels we would need source data for and will also provide you with helpful tips on how to upload and organize the files.

Further information is available in our Guide For Authors:

Revision to The EMBO Journal should be submitted online within 90 days, unless an extension has been requested and approved by the editor; please click on the link below to submit the revision online before 21st May 2025:

Link Not Available

If you choose to alternatively have this study further considered by another EMBO Press publication, please use the following hyperlink to directly transfer the manuscript, optionally with inclusion of referee reports and identities:

Link Not Available

Referee #1:

Review of the EMBO Journal manuscript 2025-120195-T entitled "A chromatin remodeling-independent role for ATRX in protecting centromeric cohesion" by Fangwei Wang and co-workers

In the manuscript at hand, Zhao et al. identify the chromatin remodeler ATRX as a direct binding partner of Pds5b. They map the interaction to the N-terminus of Pds5b, on the one hand, and to a RSYKQ motif centered around position 1419 of ATRX, on the other hand. Depletion of endogenous ATRX or its replacement with a Pds5b-binding deficient Y1419 variant results in a cohesion defect. Importantly, a small, catalytically inactive fragment of ATRX can also revert the phenotype when expressed in fusion with centromere-targeting domain arguing that this pro-cohesive function of ATRX is independent of chromatin

remodeling. Moreover, loss of cohesion in the absence of ATRX is also rescued by simultaneous depletion of Wapl. Finally, ATRX is localized to pericentromeres by spread-IF analysis. Based on these results and additional competition assays, Wang and coworkers propose that ATRX protects centromeric cohesion by competing with Wapl for the binding to Pds5b.

Overall, the ATRX-Pds5b interaction is demonstrated very convincingly and the details of this interaction is supported by high quality data. However, the observed cohesion defect is rather mild, given that the authors propose nothing less than an exchange of sororin for ATRX (and haspin) on Pds5b during metaphase, i.e. during the time between deprotection of pericentromeric cohesin (by departure of Sgo1-PP2A) until onset of separase-triggered anaphase. Below, I am therefore suggesting some straight-forward experiments that would further support the authors' provocative model. I also list some additional issues that I noted and wonder about.

Major points:

- 1) In their introduction, the authors put forward that the deprotection of pericentromeric cohesin by the departure of Sgo1-PP2A "strongly suggests that additional factors are involved in antagonizing the Wapl-Pds5 interaction at centromeric regions." Here, they fail to at least mention the alternative possibility, i.e. that the prophase pathway may no longer be active at this time, as recently proposed by Hellmuth and Stemmann (2024). Moreover, according to their model, sororin would still be inactivated by phosphorylation in metaphase, disappear from inner centromeres and be replaced by ATRX (or haspin). It would greatly strengthen the manuscript if the authors could actually show this exchange by comparative spread-IFs from cells arrested in metaphase only briefly versus for an extended period of time.
- 2) Given their previous work and expertise, the authors are in a perfect position to address whether the rather subtle cohesion defect is enhanced when they interfere with ATRX and haspin function at the same time. Again, showing this would lend further support to their provocative claim.
- 3) According to the authors' model, the ATRX-Pds5b interaction (other than the sororin-Pds5b interaction) must not be negatively regulated by mitotic phosphorylation. If true, then one would expect ATRX to spread (along with cohesin but unlike sororin) to mitotic chromosome arms when Wapl is depleted. Is that what they see? An easy experiment....
- 4) In the current form, the manuscript falls short in demonstrating that protection of centromeric cohesion by the Pds5B-binding fragment of ATRX does, in fact, require centromere-tethering. The authors need to express and analyze ATRX (1394-1443)-GFP alone, i.e. without fusion to the centromere-targeting domain of CENP-B. The competition experiments shown in figure 7 suggest that this may actually be sufficient to functionally replace endogenous ATRX.
- 5) To validate the ATRX antibody and demonstrate its specificity, the IF shown in figure 3G needs to be repeated with cells from which endogenous ATRX has been depleted.
- 6) Lagging chromosomes is not an expected phenotype of premature loss of cohesion. Instead, one typically observes a mitotic arrest due to activation of the tension-sensitive branch of the spindle assembly checkpoint. Can the authors show that cycling cells, in which ATRX is replaced by the Y1419A variant, do at least slightly accumulate in mitosis?

Minor points:

- 7) Whether DNA is loaded through the Smc1-Smc3 gate, as stated in the introduction, has been put into question by Murayama and Uhlmann, 2015, and, hence, is under debate.
- 8) The authors might also want to mention (in their introduction) that, next to aurora B and Cdk1, Plk1 and Nek2A were also reported to function in prophase pathway signaling.
- 9) Why is haspin absent in the list of Pds5b interactors identified by IP-MS?
- 10) Figure 1C is missing a loading control.
- 11) Rankin and coworkers are incorrectly cited in the discussion (3rd paragraph). They do not claim in their 2011 JBC paper that "sororin mutants lacking Pds5-binding motifs still retain the ability to protect sister-chromatid cohesion." Instead, it says in their abstract: "We also show that derivatives of sororin with deletions or mutations in the conserved C terminus fail to rescue the loss-of-cohesion phenotype caused by sororin RNAi and that these mutations also abrogate the association of sororin with the cohesin complex."
- 12) The authors should at least mention somewhere in the manuscript that it has recently been questioned whether the binding of the YSR motif to the AP[D/E]AP-loop of Pds5 is at all involved in the Wapl-mediated release of cohesin (Nasmyth et al., 2023).
- 13) The authors need to discuss whether or not the ATRX-Pds5b interaction is cell cycle regulated. For example, how is prevented that ATRX competes with Wapl in G1, when cohesin needs to be dynamic? And does ATRX (at least partly) colocalize with Pds5b in G2 phase?
- 14) Does ATRX also interact with Pds5a?

Referee #2:

Overview and general recommendation:

The authors identify ATRX as a new binding partner of cohesin. ATRX was previously known to play a role in centromere and telomere cohesion. They find that ATRX is mostly localized to the inner centromere and that ATRX depletion weakens centromeric cohesion. They also find that ATRX interacts directly with PDS5B and that this interaction is required for ATRX to protect centromeric cohesion. They identify a specific motif of ATRX around amino acid residues 1394-1443 and determine that

Y1419 is required for binding to PDS5B. Accordingly, an ATRX Y1419A mutation interferes with centromeric cohesion. Centromere tethering of the ATRX (1394-1443) fragment, but not a PDS5B-binding-deficient Y1419A version, rescues sister chromatid cohesion defects in ATRX-depleted cells. The authors further demonstrate that ATRX directly competes with WAPL for binding to PDS5B. Loss of centromeric protection upon ATRX depletion can be rescued by WAPL depletion.

This is an interesting paper with potential new insights into the mechanisms of cohesin protection at the centromere. The experiments are mostly well done and well controlled. As ATRX depletion results in a reduction, but not a complete loss, of sister chromatid cohesion, the question arises as to the relative importance of this mechanism compared to, for example, more dominant WAPL protection mechanisms exerted by SGO1-PP2A at the inner centromere. E.g. the SGO1 interaction with cohesin-SA1/2 is essential for SGO1 localization to inner centromeres and cohesin protection against WAPL release (Hara et al. 2014; Garcia-Nieto et al, 2023). They seem to suggest that ATRX could play a role in cohesin protection after the SGO1-PP2A complex redistributes from the inner centromere to kinetochore-proximal regions upon establishment of bipolar microtubule attachment. Such redistribution of SGO1 to the kinetochores is thought to result in cohesin deprotection at inner centromeres. It would be important to test this model and the importance of ATRX in cells in which SGO1 is depleted. Depending on the outcome, this reviewer would recommend that they discuss the relative importance of the different centromeric cohesin protection mechanisms against WAPL release.

Major comments:

1. Fig3: Depletion of ATRX only has a mild chromosome misalignment phenotype and only after prolonged treatment with MG132, raising questions on the importance of ATRX as compared to other well-known centromeric cohesin protectors such as SGO1. If their model is that ATRX contributes to cohesin protection at the inner centromere, they should verify whether centromeric cohesion is at least partially retained upon SGO1 depletion in ATRX wild-type cells, while such centromeric cohesion would be expected to be lost in ATRX depleted cells.
2. Fig. 2A: Expression of GST-PDS5 301-1120 is not convincing, as the protein is mostly degraded, indicative of structural/conformational heterogeneity. Probing interactions with misfolded/degraded protein is not informative. While it is reasonable to conclude that PDS5 1-300 interacts with ATRX, they can not rule out that ATRX also interacts with other PDS5 segments. This reviewer would recommend focussing on aspects that are solid and rigorous and removing data that are unconvincing.
3. Fig4D/G: Are ATRX-GFP and ATRX (Y1419A)-GFP located at centromeres? Are they depleted in cells treated with siATRX1. The current images are unclear.
4. Fig3G: Is ATRX always located at the inner centromere? They state that ATRX only occasionally locates to the inner centromere. Can they quantify from data shown in Fig3G how many centromeres contain ATRX?
5. Fig3G: Can they repeat the immunofluorescence experiments in cells expressing the ATRX (Y1419A) variant? Is ATRX (Y1419A) still localized at the inner centromere?
6. Does expression of ATRX or centromere tethering bypass SGO1-mediated cohesin protection?

Minor comment:

1. Fig3D: What is shown in the y-axis? Are these phenotypes of individual cells? How are they ranked?

Referee #3:

During mitosis, centromeric sister chromatid cohesion, which is mediated by cohesin complex, is protected by Sgo1-PP2A until metaphase/anaphase transition. It has been established that Sgo1-PP2A antagonizes the phosphorylations of cohesin and Sororin in order to protect cohesin from Wapl-dependent dissociation. However, the redistribution of Sgo1-PP2A from the inner centromere to the kinetochore well before anaphase onset suggests the involvement of an additional factor in antagonizing Wapl activity. A SWI/SNF family chromatin remodeller, ATRX, has been shown to be required for sister chromatid cohesion, though the mechanism has not yet been elucidated.

In this manuscript, Zhao and the colleagues addressed the detailed mechanism of how ATRX contributes to mitotic cohesion. They found that Pds5B directly associated with ATRX in human cells. They show that, in biochemical assays, middle region of ATRX including Y1419 and N-terminus region (1-300) of Pds5B are interacted. In HeLa cells, the authors demonstrated that ATRX was localized at the centromere region on mitotic chromosomes and that ATRX knock-down caused sister chromatid cohesion defects. ATRX-dependent cohesion requires the Y1419 residue, and, interestingly, the Pds5B-binding region (1394-1443) alone, but not its Y1419A mutant, could restore cohesion deficiency in ATRX-depleted cells. Furthermore, the authors demonstrated that cohesion defects resulting from ATRX depletion can be overcome by Wapl depletion. The biochemical analyses suggested that Wapl (1-28) and ATRX (1394-1443) competed for Pds5B, which is the reason why centromeric cohesion is protected from Wapl in mitosis.

The finding of direct binding between Pds5B and ATRX is novel, and their biochemical and cell-biological data support the hypothesis that ATRX antagonizes Wapl's function and protects cohesion in mitosis. However, several additional experiments are needed to make their hypothesis of ATRX and Wapl antagonism more rigid and also to make the function of ATRX in interphase clearer. If the following points were satisfactorily answered by authors, I would recommend the manuscript for publication in EMBO Journal.

Specific major concerns:

1) In this manuscript, authors only focused on Pds5B. How is the contribution of Pds5A in ATRX-dependent cohesion protection? If this is specific to Pds5B, what is the reason?

2) What about the relationship between Pds5, Wapl, and Sororin with ATRX in interphase? Because ATRX seems to associate with Pds5B also in interphase (Figure 7G), does this association compete with Sororin binding too?

3) Related to the previous point, if ATRX acts to antagonize Wapl also in interphase, then is ATRX required for interphase cohesion establishment like Sororin? The authors could compare the interphase cohesion by measuring sister FISH distances in G2 phase cells in the presence or absence of ATRX.

4) In the tethering experiment (Figure 5), are Pds5B and cohesin complex recruited and bound to the CB-ATRX (1394-1443), but not Y1419A?

5) If ATRX indeed protects cohesive cohesin after Sgo1-PP2A dissociation in metaphase, CB-ATRX (1394-1443) tethering should be able to maintain cohesion in the absence of Sgo1. Is this the case?

Point-by-Point response to three referees' comments:

We sincerely thank all three referees for their insightful and constructive comments, which we have now fully addressed. Please find our detailed, point-by-point responses to the reviewers' comments below (in blue):

Referee #1:

Review of the EMBO Journal manuscript 2025-120195-T entitled "A chromatin remodeling-independent role for ATRX in protecting centromeric cohesion" by Fangwei Wang and co-workers

In the manuscript at hand, Zhao et al. identify the chromatin remodeler ATRX as a direct binding partner of Pds5b. They map the interaction to the N-terminus of Pds5b, on the one hand, and to a RSYKQ motif centered around position 1419 of ATRX, on the other hand. Depletion of endogenous ATRX or its replacement with a Pds5b-binding deficient Y1419 variant results in a cohesion defect. Importantly, a small, catalytically inactive fragment of ATRX can also revert the phenotype when expressed in fusion with centromere-targeting domain arguing that this pro-cohesive function of ATRX is independent of chromatin remodeling. Moreover, loss of cohesion in the absence of ATRX is also rescued by simultaneous depletion of Wapl. Finally, ATRX is localized to pericentromeres by spread-IF analysis. Based on these results and additional competition assays, Wang and coworkers propose that ATRX protects centromeric cohesion by competing with Wapl for the binding to Pds5b.

Overall, the ATRX-Pds5b interaction is demonstrated very convincingly and the details of this interaction is supported by high quality data. However, the observed cohesion defect is rather mild, given that the authors propose nothing less than an exchange of sororin for ATRX (and haspin) on Pds5b during metaphase, i.e. during the time between deprotection of pericentromeric cohesin (by departure of Sgo1-PP2A) until onset of separase-triggered anaphase. Below, I am therefore suggesting some straight-forward experiments that would further support the authors' provocative model. I also list some additional issues that I noted and wonder about.

Reply: We appreciate the reviewer's constructive suggestions and have considered these additional experiments and discussions, which further support our model, in the revised manuscript and in our discussion of future directions.

Major points:

1) In their introduction, the authors put forward that the deprotection of pericentromeric cohesin by the departure of Sgo1-PP2A "strongly suggests that additional factors are involved in antagonizing the Wapl-Pds5 interaction at centromeric regions."

Here, they fail to at least mention the alternative possibility, i.e. that the prophase pathway may no longer be active at this time, as recently proposed by Hellmuth and Stemmann (2024).

Reply: We thank the reviewer for pointing this out. In the revised manuscript, we now state in the Introduction: “This suggests that either Wapl becomes globally inactive during metaphase (Hellmuth & Stemmann, 2024), or that additional factors are involved in antagonizing the Wapl-Pds5 interaction specifically at mitotic centromeres.”

Moreover, according to their model, sororin would still be inactivated by phosphorylation in metaphase, disappear from inner centromeres and be replaced by ATRX (or haspin). It would greatly strengthen the manuscript if the authors could actually show this exchange by comparative spread-IFs from cells arrested in metaphase only briefly versus for an extended period of time.

Reply: We thank the referee for raising this question. Following the suggestion, we performed new experiments. HeLa cells and HeLa cells stably expressing Sororin-GFP were treated with nocodazole for 3 hours or with MG132 for 1, 3, and 6 hours. Mitotic chromosome spreads were then stained for ATRX, GFP, ACA, and DAPI, and we quantified the fluorescence intensity ratios of ATRX and Sororin-GFP to ACA. Our new results show that ATRX’s centromeric localization remains stable during prolonged metaphase arrest (Fig. EV4D, E; **new data**), while Sororin-GFP is gradually displaced from centromeres (Fig. EV4F-H; **new data**). These findings support our hypothesis that phosphorylation may inactivate Sororin during metaphase, leading to its displacement from inner centromeres, where ATRX then predominates.

2) Given their previous work and expertise, the authors are in a perfect position to address whether the rather subtle cohesion defect is enhanced when they interfere with ATRX and haspin function at the same time. Again, showing this would lend further support to their provocative claim.

Reply: We appreciate the reviewer’s suggestion. Evaluating the combined effect of ATRX and Haspin interference on sister chromatid cohesion is indeed informative. Our results already demonstrate that ATRX strengthens centromeric cohesion by binding to Pds5B via a conserved RSYK-[Q/K] motif, while previous work (Goto et al., 2017; Zhou et al., 2017) has shown that Haspin promotes cohesion through a similar [R/K]-TYG-[R/K] motif-mediated interaction with Pds5B. To address the reviewer’s point, we assessed the impact of ATRX knockdown on sister chromatid cohesion in HeLa cells with CRISPR/Cas9-mediated Haspin knockout (KO).

Our new data show that Haspin KO cells exhibit defects in maintaining sister chromatid cohesion during metaphase arrest induced by MG132 treatment for 2-6 h (Fig. 9A-C; **new data**). Notably, ATRX knockdown further exacerbates cohesion loss in Haspin KO cells, indicating that ATRX contributes to sister chromatid cohesion independently of Haspin. ATRX depletion also aggravates metaphase chromosome alignment defects in Haspin KO cells (Fig. 9D; **new data**). Thus, ATRX and Haspin play additive roles in protecting centromeric cohesion.

Given the similarity in their Pds5B-binding motifs, we further examined the effect of centromere tethering of the Pds5B-binding fragment of ATRX in Haspin KO cells. Our new results

demonstrate that expression of CB-ATR_X (1394-1443)-GFP, but not CB-GFP or CB-ATR_X (1394-1443)-Y1419A-GFP, mitigate the loss of metaphase cohesion in Haspin KO cells (Appendix Fig. S3A-C; **new data**), suggesting that ATR_X and Haspin promote centromeric cohesion through at least partially similar mechanisms.

Consequently, in the revised Discussion, we state: “We propose that at least two functional pools of cohesin exist at mitotic centromeres: one stabilized by ATR_X-bound Pds5B and another regulated by Haspin-bound Pds5B (Fig. 9F). This partitioning of regulatory factors may provide redundancy to ensure robust centromeric cohesion under varying cellular conditions.”

3) According to the authors' model, the ATR_X-Pds5b interaction (other than the sororin-Pds5b interaction) must not be negatively regulated by mitotic phosphorylation. If true, then one would expect ATR_X to spread (along with cohesin but unlike sororin) to mitotic chromosome arms when Wapl is depleted. Is that what they see? An easy experiment....

Reply: We have shown that stably expressed Myc-Pds5B co-immunoprecipitated both Wapl and ATR_X in cells synchronized in either G₂ phase or mitosis (Fig. 8I). Following the referee's suggestion, we performed immunofluorescence microscopy and observed that ATR_X remains enriched at metaphase centromeres in Wapl-depleted cells (Fig. 7H, I; **new data**). This finding is not surprising since ATR_X's mitotic centromeric localization is HP1-dependent (Fig. 4B; **new data**), and depletion of Wapl is not expected to impact HP1 localization at centromeres.

4) In the current form, the manuscript falls short in demonstrating that protection of centromeric cohesion by the Pds5B-binding fragment of ATR_X does, in fact, require centromere-tethering. The authors need to express and analyze ATR_X (1394-1443)-GFP alone, i.e. without fusion to the centromere-targeting domain of CENP-B. The competition experiments shown in figure 7 suggest that this may actually be sufficient to functionally replace endogenous ATR_X.

Reply: We agree with the reviewer regarding the necessity of demonstrating the requirement for centromere-tethering. To address this, HeLa cells were transfected with control or ATR_X siRNA along with plasmids encoding either CB-ATR_X (1394-1443)-GFP or untethered ATR_X (1394-1443)-GFP. Following an 8-hour MG132 treatment, we prepared mitotic chromosome spreads, stained with DAPI, and quantified the percentage of cells showing predominantly separated versus unseparated sister chromatids. Our data indicate that expression of the untethered ATR_X (1394-1443)-GFP fragment fails to rescue the cohesion defect following ATR_X knockdown (Appendix Fig. S1C-E; **new data**). This result is expected because the untethered fragment lacks the HP1-binding motif necessary for centromere localization, and thus it cannot antagonize Wapl binding to Pds5B at mitotic centromeres. This new finding reinforces our conclusion that centromere-tethering of the Pds5B-binding fragment of ATR_X is essential for the functional replacement of endogenous ATR_X in protecting centromeric cohesion.

5) To validate the ATR_X antibody and demonstrate its specificity, the IF shown in figure 3G needs to be repeated with cells from which endogenous ATR_X has been depleted.

Reply: We thank the reviewer for this important suggestion. To validate the specificity of the ATRX antibody, HeLa cells were transfected with either control or ATRX siRNA. Forty-eight hours post-transfection, cells were treated with MG132 for 3 hours, and mitotic chromosome spreads were prepared and stained for ATRX, ACA, and DAPI. Super-resolution microscopy reveals that ATRX predominantly localizes to the inner centromere, with 31.6% of metaphase chromosomes exhibiting at least two ATRX foci at bipartite centromere subdomains that resemble the cohesin-accumulating regions (Ruiz et al., 2024; Sacristan et al., 2024; Sen Gupta et al., 2023) (Fig. 4C, D; **new data**). Importantly, these foci disappeared in cells transfected with ATRX siRNA, thereby validating the specificity of the ATRX signal.

6) Lagging chromosomes is not an expected phenotype of premature loss of cohesion. Instead, one typically observes a mitotic arrest due to activation of the tension-sensitive branch of the spindle assembly checkpoint. Can the authors show that cycling cells, in which ATRX is replaced by the Y1419A variant, do at least slightly accumulate in mitosis?

Reply:

We thank the reviewer for this insightful comment. We agree that compromised sister chromatid cohesion typically activates the spindle assembly checkpoint (SAC), leading to mitotic arrest, whereas our data show an increased frequency of anaphase lagging chromosomes. We interpret this discrepancy as evidence of a relatively mild cohesion defect that allows cells to proceed into anaphase but results in merotelic kinetochore-microtubule (KT-MT) attachments and lagging chromosomes. This interpretation is supported by recent findings (Sacristan et al., 2024) showing that cohesin helps stabilize bipartite centromere subdomains, thereby preventing erroneous KT-MT attachments. Consistent with this, our super-resolution microscopy reveals ATRX localization at these subdomains (Fig. 4C, D, G, H; **new data**), suggesting that ATRX-bound cohesin contributes to maintaining centromeric subdomain integrity.

To directly address the reviewer's question, we performed a mitotic index analysis in cycling HeLa cells and in HeLa cells stably expressing ATRX-GFP (wild-type or Y1419A), following transfection with control or ATRX siRNA. Forty-eight hours post-transfection, ATRX knockdown increases the mitotic index by 2.2-fold in control HeLa cells and by 2.6-fold in ATRX (Y1419A)-GFP-expressing cells, while no significant change is observed in cells expressing ATRX-GFP (WT) (Fig. EV5D, E; **new data**). These results indicate that cells expressing the Y1419A mutant, which disrupts the ATRX-Pds5B interaction, do indeed exhibit a modest accumulation in mitosis, consistent with partial SAC activation in response to mild cohesion defects.

Minor points:

7) Whether DNA is loaded through the Smc1-Smc3 gate, as stated in the introduction, has been put into question by Murayama and Uhlmann, 2015, and, hence, is under debate.

Reply: We thank the reviewer for pointing out this important clarification. In the revised Introduction, we now state: "In vertebrate cells, cohesin is loaded onto chromatin during mitotic telophase and early G1 phase, a process that may involve the opening of the interface between

the hinge domains of SMC1A and SMC3 (Buheitel & Stemmann, 2013; Gruber et al, 2006; Murayama & Uhlmann, 2015), although the exact mechanism requires further investigation.”

8) The authors might also want to mention (in their introduction) that, next to aurora B and Cdk1, Plk1 and Nek2A were also reported to function in prophase pathway signaling.

Reply: We thank the reviewer for this valuable suggestion. In the revised Introduction, we now state: “Starting in prophase, phosphorylation of Sororin by the mitotic kinases Aurora B and Cdk1 (Dreier et al, 2011; Liu et al, 2013b; Nishiyama et al, 2013), along with phosphorylation of Pds5 by Nek2a and cyclin A2-Cdk1/2 (Hellmuth & Stemmann, 2024), displaces Sororin from Pds5 and allows Wapl to bind Pds5, resulting in the release of most cohesin from chromosome arms. Additionally, phosphorylation of SA2 by Plk1 facilitates Wapl-mediated removal of cohesin from chromosome arms (Hauf et al, 2005; Nishiyama et al., 2013).”

9) Why is haspin absent in the list of Pds5b interactors identified by IP-MS?

Reply: We appreciate the reviewer’s observation. Northern blot analysis in human and mouse tissues have shown that Haspin is abundantly expressed in testis, while its expression is lower in various somatic tissues with high numbers of dividing cells, as well as in all proliferating cell lines tested (Higgins, 2001, PMID: 11311556; Higgins, 2003, PMID: 12737306). Consistently, to the best of our knowledge, Haspin has not been detected in any mass spectrometry-based analyses.

In the revised Results, we now state: “As expected, Myc-Pds5B co-immunoprecipitated SMC3, SMC1A, RAD21, SA2, SA1, as well as Wapl and Sororin (Fig. 1A; Dataset EV1), confirming known interactions. Haspin, another Pds5B interactor (Goto et al, 2017; Zhou et al, 2017), was not detected, possibly due to its low expression in somatic cells (Higgins, 2003).”

10) Figure 1C is missing a loading control.

Reply: We thank the reviewer for pointing this out. We have now included GAPDH as a loading control in Figure 1C.

11) Rankin and coworkers are incorrectly cited in the discussion (3rd paragraph). They do not claim in their 2011 JBC paper that "sororin mutants lacking Pds5-binding motifs still retain the ability to protect sister-chromatid cohesion." Instead, it says in their abstract: "We also show that derivatives of sororin with deletions or mutations in the conserved C terminus fail to rescue the loss-of-cohesion phenotype caused by sororin RNAi and that these mutations also abrogate the association of sororin with the cohesin complex."

Reply: We thank the reviewer for pointing this out. Upon revisiting the study by Rankin and coworkers, we note that they reported: “Mutations within the other conserved regions of the sororin protein, such as a deletion of region A or point mutations in B, did not have strong impact on cohesion activity (Δ A and B*)” (see the Figure 4A in Wu et al., 2011). Notably, region A

comprises the residues KVRRSYSR.

In light of this, we have revised the Discussion to state: “Our discovery of the ATRX-Pds5B interaction may also explain the interesting observation that deletion of the YSR motif-containing residues KVRRSYSRL of Sororin does not strongly affect cohesion (Wu et al, 2011).”

12) The authors should at least mention somewhere in the manuscript that it has recently been questioned whether the binding of the YSR motif to the AP[D/E]AP-loop of Pds5 is at all involved in the Wapl-mediated release of cohesin (Nasmyth et al., 2023).

Reply: We thank the reviewer for pointing this out. Kim Nasmyth noted that “Wapl’s N- terminal YSR has never been rigorously demonstrated to be necessary for its RA”, raising the possibility that the YSR motif may not be essential for Wapl-mediated cohesin release.

However, experimental evidence from Ouyang et al. (2016) showed that mutating the YSR motif (YSR to ASE) in human Wapl impairs, but does not abolish, its ability to rescue the chromosome arm resolution defect caused by Wapl depletion. This suggests that while the YSR motif is not strictly essential, it promotes Wapl-mediated cohesin release (Ouyang et al., Mol Cell, 2016, PMID: 26971492).

We have revised the Discussion to state: “Although the N-terminal YSR motif may not be essential for the cohesin release activity of Wapl (Nasmyth et al, 2023), this motif clearly promotes Wapl-mediated resolution of chromosome arms in early mitosis (Ouyang et al., 2016)”.

13) The authors need to discuss whether or not the ATRX-Pds5b interaction is cell cycle regulated. For example, how is prevented that ATRX competes with Wapl in G1, when cohesin needs to be dynamic? And does ATRX (at least partly) colocalize with Pds5b in G2 phase?

Reply: We thank the reviewer for raising these points. Regarding whether the ATRX-Pds5B interaction is cell cycle regulated, we have shown in the manuscript that stably expressed Myc-Pds5B co-immunoprecipitated ATRX in cells synchronized in either G2 phase or mitosis, and that the amount of co-immunoprecipitated ATRX was slightly higher in mitotic cells compared to G2-phase cells (Fig. 8I).

Regarding how ATRX is prevented from competing with Wapl in G1, when cohesin needs to remain dynamic, our new results show that ATRX protein levels gradually increase from G1 to S phase (Appendix Fig. S2A; **new data**). In addition, Myc-Pds5B co-immunoprecipitates Wapl much more efficiently than ATRX in G1-phase cells (Appendix Fig. S2B; **new data**). Furthermore, ATRX knockdown does not affect the interaction between Myc-Pds5B and Wapl in G1-phase cells (Appendix Fig. S2B; **new data**), nor the binding of Myc-Pds5B to Wapl in asynchronous cells (Appendix Fig. S2C; **new data**). These observations suggest that ATRX does not interfere with Pds5B binding to Wapl in interphase cells.

Regarding whether ATRX colocalizes with Pds5B in G2 phase, immunofluorescence microscopy demonstrates that ATRX and Pds5B localize exclusively to G2-phase nuclei (Fig. EV4A; **new data**), that ATRX colocalize with HP1 α at G2-phase heterochromatin, and this colocalization

disappears in HP1 α and HP1 γ double knockout (DKO) cells (Fig. EV4B, C; **new data**). These findings align with the established role of HP1-ATRAX interaction in recruiting ATRAX to heterochromatin (Eustermann et al, 2011; Iwase et al, 2011).

14) Does ATRAX also interact with Pds5a?

Reply: We thank the reviewer for raising this interesting point. Our new observations from additional experiments show that, in contrast to Myc-Pds5B, Myc-Pds5A is not noticeably associated with either endogenous ATRAX or exogenously expressed ATRAX-GFP in co-immunoprecipitation assays (Fig. EV1B, C; **new data**).

Accordingly, we have revised the Discussion to state: “We noticed that, in co-immunoprecipitation assays using HeLa cell lysates, ATRAX associated with Pds5B but was not detectably bound to Pds5A (Fig. EV1B, C), suggesting that ATRAX preferentially interacts with Pds5B. This preference may be due to differences in intracellular localization, post-translational modifications, or the distinct unstructured C-terminal domains of Pds5B and Pds5A (Carretero et al., 2013; Hellmuth & Stemmann, 2024; Losada et al., 2005). The preferential binding of Pds5B to ATRAX may also reflect the specific role for Pds5B, but not Pds5A, in protecting centromeric cohesion (Carretero et al, 2013). Future studies will be needed to clarify these possibilities.”

Referee #2:

Overview and general recommendation:

The authors identify ATRAX as a new binding partner of cohesin. ATRAX was previously known to play a role in centromere and telomere cohesion. They find that ATRAX is mostly localized to the inner centromere and that ATRAX depletion weakens centromeric cohesion. They also find that ATRAX interacts directly with PDS5B and that this interaction is required for ATRAX to protect centromeric cohesion. They identify a specific motif of ATRAX around amino acid residues 1394-1443 and determine that Y1419 is required for binding to PDS5B. Accordingly, an ATRAX Y1419A mutation interferes with centromeric cohesion. Centromere tethering of the ATRAX (1394-1443) fragment, but not a PDS5B-binding-deficient Y1419A version, rescues sister chromatid cohesion defects in ATRAX-depleted cells. The authors further demonstrate that ATRAX directly competes with WAPL for binding to PDS5B. Loss of centromeric protection upon ATRAX depletion can be rescued by WAPL depletion.

This is an interesting paper with potential new insights into the mechanisms of cohesin protection at the centromere. The experiments are mostly well done and well controlled. As ATRAX depletion results in a reduction, but not a complete loss, of sister chromatid cohesion, the question arises as to the relative importance of this mechanism compared to, for example, more dominant WAPL protection mechanisms exerted by SGO1-PP2A at the inner centromere. E.g. the SGO1 interaction with cohesin-SA1/2 is essential for SGO1 localization to inner centromeres and cohesin protection against WAPL release (Hara et al. 2014; Garcia-Nieto et al, 2023). They seem to suggest that ATRAX could play a role in cohesin protection after the SGO1-PP2A complex redistributes from the inner centromere to kinetochore-proximal regions upon

establishment of bipolar microtubule attachment. Such redistribution of SGO1 to the kinetochores is thought to result in cohesin deprotection at inner centromeres. It would be important to test this model and the importance of ATRX in cells in which SGO1 is depleted. Depending on the outcome, this reviewer would recommend that they discuss the relative importance of the different centromeric cohesin protection mechanisms against WAPL release.

Reply: We appreciate the reviewer's helpful comments, which have strengthened both the experimental design and the interpretation of our findings.

Major comments:

1. Fig3: Depletion of ATRX only has a mild chromosome misalignment phenotype and only after prolonged treatment with MG132, raising questions on the importance of ATRX as compared to other well-known centromeric cohesin protectors such as SGO1. If their model is that ATRX contributes to cohesin protection at the inner centromere, they should verify whether centromeric cohesion is at least partially retained upon SGO1 depletion in ATRX wild-type cells, while such centromeric cohesion would be expected to be lost in ATRX depleted cells.

Reply: We thank the reviewer for this insightful and constructive comment. We agree that it is important to clarify the relative contributions of different centromeric cohesin protection mechanisms, particularly in the context of Wapl antagonism by the Sgo1-PP2A complex and ATRX.

Sgo1 is a well-established protector of sister chromatid cohesion (Kitajima et al, 2004; McGuinness et al., 2005; Salic et al, 2004; Tang et al, 2004). During early mitosis, Sgo1 localizes along chromosome arms (Chu et al, 2020; Gimenez-Abian et al, 2004; Nakajima et al, 2007; Yan et al., 2024), and becomes enriched at centromeres (Kitajima et al, 2005; McGuinness et al., 2005).

Following the reviewer's suggestion, we performed additional experiments to assess whether ATRX plays a role in cohesin protection in the absence of Sgo1. Our new results show that Sgo1 knockdown led to a strong loss of sister chromatid cohesion in nocodazole-arrested mitotic cells, which was more severe than the defect caused by ATRX depletion alone (Appendix Fig. S3D-F; **new data**), indicating that Sgo1 plays the dominant role, particularly during early mitosis prior to metaphase. Moreover, combined depletion of ATRX and Sgo1 resulted in an additive cohesion defect, highlighting their non-redundant contributions.

We have also expanded the Discussion section to address this point, now stating: "While ATRX contributes to the stabilization of centromeric cohesion, its depletion results in a milder cohesion defect compared to Sgo1 depletion. This is not unexpected, as ATRX specifically reinforces centromeric cohesion, whereas Sgo1 protects cohesin both on chromosome arms and at centromeres. Our data further suggest that ATRX may become particularly important for maintaining centromeric cohesion during metaphase, after the Sgo1-PP2A complex redistributes toward kinetochore-proximal regions, a step believed to expose cohesin at the inner centromere to Wapl-mediated release. Future work will be required to dissect this temporal regulation and fully elucidate the interplay between Sgo1-PP2A, ATRX, and Wapl in the protection of centromeric cohesion".

It should also be noted that although ATRX plays a less prominent role than Sgo1 in safeguarding sister chromatid cohesion under physiological conditions, it is possible that ATRX mutations disrupting its interaction with Pds5B, which may only mildly activate the spindle assembly checkpoint and thus allow cell survival, could contribute to chromosomal instability in tumors. In contrast, Sgo1 mutations that impair cohesin interaction are more likely to strongly activate the spindle assembly checkpoint and induce mitotic catastrophe and cell death. We have therefore included the following statement in the Discussion: “Future studies should investigate whether mutations in ATRX disrupt its interaction with Pds5B and how such changes might contribute to chromosomal instability in cancer.”

2. Fig. 2A: Expression of GST-PDS5 301-1120 is not convincing, as the protein is mostly degraded, indicative of structural/conformational heterogeneity. Probing interactions with misfolded/degraded protein is not informative. While it is reasonable to conclude that PDS5 1-300 interacts with ATRX, they can not rule out that ATRX also interacts with other PDS5 segments. This reviewer would recommend focussing on aspects that are solid and rigorous and removing data that are unconvincing.

Reply: We agree with the reviewer’s comment and appreciate the suggestion. In response, we have omitted the pull-down data using GST-Pds5B fragments 301-1120 and 1121-1447 from the revised figure (now shown in Fig. EV2A). Additionally, we conducted co-immunoprecipitation assays, which demonstrate that ATRX-GFP co-immunoprecipitates the Flag-tagged Pds5B (1-300) fragment, but not with the 301-1120 or 1121-1447 fragments (Fig. 2A; **new data**). These results confirm that the interaction is specific to the N-terminal region of Pds5B.

3. Fig4D/G: Are ATRX-GFP and ATRX (Y1419A)-GFP located at centromeres? Are they depleted in cells treated with siATRX1. The current images are unclear.

Reply: We thank the reviewer for pointing this out. To address this, we re-examined the localization of ATRX-GFP and ATRX (Y1419A)-GFP in HeLa cells following siRNA-mediated ATRX depletion. HeLa cells and HeLa cells stably expressing ATRX-GFP (WT or Y1419A) were transfected with either control siRNA or ATRX siRNA. Forty-eight hours post-transfection, cells were treated with MG132 for 8 hours or nocodazole for 3 hours, and mitotic chromosome spreads were stained for GFP, ACA, and DAPI. Fluorescence microscopy reveals that both ATRX-GFP and ATRX (Y1419A)-GFP localize to mitotic centromeres in cells transfected with either control or ATRX siRNA (Fig. 5B, E; **new data**).

4. Fig3G: Is ATRX always located at the inner centromere? They state that ATRX only occasionally locates to the inner centromere. Can they quantify from data shown in Fig3G how many centromeres contain ATRX?

Reply: We appreciate the reviewer’s helpful suggestion. To address this point, we quantified ATRX localization at centromeres using super-resolution microscopy. HeLa cells were transfected with either control or ATRX siRNA, and 48 hours post-transfection, cells were treated with MG132 for 3 hours. Mitotic chromosome spreads were then stained for ATRX, ACA,

and DAPI. Our analysis show that ATRX predominantly localize to the inner centromere, with 31.6% of metaphase chromosomes displaying at least two ATRX foci at the bipartite centromere subdomains (Fig. 4C, D; **new data**), regions that closely resemble the cohesin-accumulating sites, as reported in recent studies (Ruiz et al., 2024; Sacristan et al., 2024; Sen Gupta et al., 2023). These ATRX foci were absent upon ATRX depletion, confirming the specificity of the signal.

Besides, we would also like to clarify that the term “inner centromere” is not strictly equivalent to “bipartite centromere subdomains,” and our description in the text reflects this distinction.

5. Fig3G: Can they repeat the immunofluorescence experiments in cells expressing the ATRX (Y1419A) variant? Is ATRX (Y1419A) still localized at the inner centromere?

Reply: We appreciate the reviewer’s valuable suggestion. To address this point, we performed super-resolution immunofluorescence microscopy on HeLa cells stably expressing either ATRX-GFP or ATRX (Y1419A)-GFP. Cells were treated with MG132 for 3 hours, and mitotic chromosome spreads were stained with antibodies against GFP and ACA, along with DAPI. Our results show that ATRX (Y1419A)-GFP, similar to ATRX-GFP, predominantly localizes to the inner centromere (Fig. 4G, H; **new data**). These findings indicate that the Y1419A mutation does not impair the centromeric localization of ATRX.

6. Does expression of ATRX or centromere tethering bypass SGO1-mediated cohesin protection?

Reply: We thank the reviewer for this insightful question. To address this, we examined whether centromere tethering of the Pds5B-binding ATRX fragment could compensate for the loss of Sgo1-mediated cohesin protection. Our observations show that expression of CB-ATR_X (1394-1443)-GFP does not noticeably rescue the cohesion defect in Sgo1-depleted cells (Appendix Fig. S3G-I; **new data**), consistent with the distinct mechanisms by which ATRX and Sgo1 interact with, and protect, the cohesin complex.

Minor comment:

1. Fig3D: What is shown in the y-axis? Are these phenotypes of individual cells? How are they ranked?

Reply: We thank the reviewer for pointing this out. In the revised Figure 3D, the y-axis now clearly indicates that it represents individual mitotic cells. The cells are ranked based on the duration each cell spent in prometaphase and in a pseudo-metaphase state with scattered chromosomes.

Referee #3:

During mitosis, centromeric sister chromatid cohesion, which is mediated by cohesin complex, is protected by Sgo1-PP2A until metaphase/anaphase transition. It has been established that Sgo1-PP2A antagonizes the phosphorylations of cohesin and Sororin in order to protect cohesin

from Wapl-dependent dissociation. However, the redistribution of Sgo1-PP2A from the inner centromere to the kinetochore well before anaphase onset suggests the involvement of an additional factor in antagonizing Wapl activity. A SWI/SNF family chromatin remodeller, ATRX, has been shown to be required for sister chromatid cohesion, though the mechanism has not yet been elucidated.

In this manuscript, Zhao and the colleagues addressed the detailed mechanism of how ATRX contributes to mitotic cohesion. They found that Pds5B directly associated with ATRX in human cells. They show that, in biochemical assays, middle region of ATRX including Y1419 and N-terminus region (1-300) of Pds5B are interacted. In HeLa cells, the authors demonstrated that ATRX was localized at the centromere region on mitotic chromosomes and that ATRX knock-down caused sister chromatid cohesion defects. ATRX-dependent cohesion requires the Y1419 residue, and, interestingly, the Pds5B-binding region (1394-1443) alone, but not its Y1419A mutant, could restore cohesion deficiency in ATRX-depleted cells. Furthermore, the authors demonstrated that cohesion defects resulting from ATRX depletion can be overcome by Wapl depletion. The biochemical analyses suggested that Wapl (1-28) and ATRX (1394-1443) competed for Pds5B, which is the reason why centromeric cohesion is protected from Wapl in mitosis.

The finding of direct binding between Pds5B and ATRX is novel, and their biochemical and cell-biological data support the hypothesis that ATRX antagonizes Wapl's function and protects cohesion in mitosis. However, several additional experiments are needed to make their hypothesis of ATRX and Wapl antagonism more rigid and also to make the function of ATRX in interphase clearer. If the following points were satisfactorily answered by authors, I would recommend the manuscript for publication in EMBO Journal.

Reply: We sincerely thank the reviewer for the thorough evaluation of our manuscript and for recognizing the novelty and significance of our findings.

Specific major concerns:

1) In this manuscript, authors only focused on Pds5B. How is the contribution of Pds5A in ATRX-dependent cohesion protection? If this is specific to Pds5B, what is the reason?

Reply: We thank the reviewer for raising this interesting point. Our new observations from additional experiments show that, in contrast to Myc-Pds5B, Myc-Pds5A is not noticeably associated with either endogenous ATRX or exogenously expressed ATRX-GFP in co-immunoprecipitation assays (Fig. EV1B, C; **new data**).

Accordingly, we have revised the Discussion to state: “We noticed that, in co-immunoprecipitation assays using HeLa cell lysates, ATRX associated with Pds5B but was not detectably bound to Pds5A (Fig. EV1B, C), suggesting that ATRX preferentially interacts with Pds5B. This preference may be due to differences in intracellular localization, post-translational modifications, or the distinct unstructured C-terminal domains of Pds5B and Pds5A (Carretero et al., 2013; Hellmuth & Stemmann, 2024; Losada et al., 2005). The preferential binding of Pds5B to ATRX may also reflect the specific role for Pds5B, but not Pds5A, in protecting centromeric cohesion (Carretero et al, 2013). Future studies will be needed

to clarify these possibilities.”

2) What about the relationship between Pds5, Wapl, and Sororin with ATRX in interphase? Because ATRX seems to associate with Pds5B also in interphase (Figure 7G), does this association compete with Sororin binding too?

Reply: We thank the reviewer for raising these thoughtful points.

Regarding the relationship between Pds5B, Sororin, and ATRX in interphase, we have shown in the manuscript that stably expressed Myc-Pds5B co-immunoprecipitates both ATRX and Sororin in G2-phase cells, with the co-immunoprecipitation efficiency of ATRX being notably lower than that of Sororin (Fig. 8I). Importantly, our new results show that ATRX knockdown does not affect the interaction between Myc-Pds5B and Sororin in asynchronous cells (Appendix Fig. S2C; **new data**). Moreover, Sororin contains two conserved motifs, the FGF motif (Nishiyama et al., 2010, Cell, PMID: 21111234) and the YSR motif (Ouyang et al., 2016, Mol Cell, PMID: 26971492), both of which are required for its interaction with Pds5B in human HeLa cells (see Figure 1E in Ouyang et al., 2016.). Taken together, these findings strongly suggest that the ATRX does not interfere with the Pds5B-Sororin interaction in interphase cells.

Regarding the relationship between Pds5B, Wapl, and ATRX in interphase, we have shown that stably expressed Myc-Pds5B co-immunoprecipitates both ATRX and Wapl in G2-phase cells, although the amount of co-immunoprecipitated ATRX is much less than Wapl (Fig. 8I). Our new observations further indicate that ATRX protein levels gradually increase from G1 to S phase (Appendix Fig. S2A; **new data**). Moreover, Myc-Pds5B co-immunoprecipitates Wapl much more efficiently than ATRX in G1-phase cells (Appendix Fig. S2B; **new data**). Additionally, ATRX knockdown does not affect the interaction between Myc-Pds5B and Wapl in either G1-phase cells (Appendix Fig. S2B; **new data**) or asynchronous cells (Appendix Fig. S2C; **new data**). These observations strongly suggest that the ATRX does not affect the Pds5B-Wapl interaction in interphase cells.

We state in the Discussion that “ATRX has been shown to colocalize with cohesin at specific genomic loci, and its loss reduces cohesin occupancy at these sites (Kernohan et al, 2010). These observations raise the possibility that ATRX-bound cohesin may also play a role in modulating DNA looping. Whether ATRX’s interaction with Pds5 influences cohesin dynamics in interphase remains an open question, warranting further investigation into its role in chromatin organization.”

3) Related to the previous point, if ATRX acts to antagonize Wapl also in interphase, then is ATRX required for interphase cohesion establishment like Sororin? The authors could compare the interphase cohesion by measuring sister FISH distances in G2 phase cells in the presence or absence of ATRX.

Reply: We thank the reviewer for this thoughtful suggestion. To address whether ATRX is required for the establishment of sister chromatid cohesion during interphase, we examined cohesion in ATRX-depleted cells. Specifically, we used EGFP-dCas9 together with a sgRNA to label endogenous genomic loci on the arm of chromosome 15 in HeLa cells (Chen et al., 2013;

Chen et al., 2018). We then measured the distance between the fluorescently labeled sister loci (“doublets”) in cells synchronized in G2 phase. While Sororin knockdown leads to a significant increase in the distance between sister loci, depletion of ATRX does not show any measurable effect (Fig. EV3H-L; **new data**). These results indicate that ATRX is not required for the establishment of sister chromatid cohesion during interphase, further supporting the notion that ATRX does not interfere with the Pds5B-Wapl interaction in G2-phase cells.

4) In the tethering experiment (Figure 5), are Pds5B and cohesin complex recruited and bound to the CB-ATRX (1394-1443), but not Y1419A?

Reply: We thank the reviewer for this insightful question. To examine whether Pds5B is recruited by CB-ATRX (1394-1443), we transfected HeLa cells stably expressing Myc-Pds5B with plasmids encoding CB-GFP, CB-ATRX (1394-1443)-GFP, or CB-fused RAD21-GFP (used as a positive control). Twenty-four hours post-transfection, cells were treated with nocodazole for 3 hours, and mitotic chromosome spreads were stained for the Myc tag and DAPI. Immunofluorescence microscopy reveals that CB-ATRX (1394-1443)-GFP only minimally recruits Pds5B to mitotic centromeres, in clear contrast to the robust recruitment observed with CB-RAD21-GFP (Appendix Fig. S1A; **new data**).

To assess whether the cohesin complex is recruited by CB-ATRX (1394-1443), we performed a similar experiment using cells transfected with CB-GFP, CB-ATRX (1394-1443)-GFP, or CB-RAD21-GFP, followed by nocodazole treatment and immunostaining for SA2 and DAPI. Consistent with the Pds5B results, CB-ATRX (1394-1443)-GFP barely recruit SA2 to mitotic centromeres, again in contrast to CB-RAD21-GFP (Appendix Fig. S1B; **new data**).

These observations suggest that CB-ATRX (1394-1443)-GFP does not promote cohesion by actively recruiting Pds5B or cohesin to centromeres. Instead, it likely protects centromeric cohesion by binding to pre-existing, Pds5B-associated cohesin complexes at mitotic centromeres.

5) If ATRX indeed protects cohesive cohesin after Sgo1-PP2A dissociation in metaphase, CB-ATRX (1394-1443) tethering should be able to maintain cohesion in the absence of Sgo1. Is this the case?

Reply: We thank the reviewer for this insightful question. To address this, we examined whether centromere tethering of the Pds5B-binding ATRX fragment could compensate for the loss of Sgo1-mediated cohesin protection. Our results show that expression of CB-ATRX (1394-1443)-GFP does not noticeably rescue the cohesion defect in Sgo1-depleted cells (Appendix Fig. S3G-I; **new data**), consistent with the distinct mechanisms by which ATRX and Sgo1 interact with, and protect, the cohesin complex.

Sgo1 is a well-established protector of sister chromatid cohesion (Kitajima et al, 2004; McGuinness et al., 2005; Salic et al, 2004; Tang et al, 2004). During early mitosis, Sgo1 localizes along chromosome arms (Chu et al, 2020; Gimenez-Abian et al, 2004; Nakajima et al, 2007; Yan et al., 2024), and becomes enriched at centromeres (Kitajima et al, 2005; McGuinness et al., 2005).

We performed additional experiments to assess whether ATRX plays a role in cohesin protection in the absence of Sgo1. Our new results show that Sgo1 knockdown leads to a strong loss of sister chromatid cohesion in nocodazole-arrested mitotic cells, which is more severe than the defect caused by ATRX depletion alone (Appendix Fig. S3D-F; **new data**), indicating that Sgo1 plays the dominant role, particularly during early mitosis prior to metaphase. Moreover, combined depletion of ATRX and Sgo1 resulted in an additive cohesion defect, highlighting their non-redundant contributions.

We have also expanded the Discussion section to address this point, now stating: “While ATRX contributes to the stabilization of centromeric cohesion, its depletion results in a milder cohesion defect compared to Sgo1 depletion. This is not unexpected, as ATRX specifically reinforces centromeric cohesion, whereas Sgo1 protects cohesin both on chromosome arms and at centromeres. Our data further suggest that ATRX may become particularly important for maintaining centromeric cohesion during metaphase, after the Sgo1-PP2A complex redistributes toward kinetochore-proximal regions, a step believed to expose cohesin at the inner centromere to Wapl-mediated release. Future work will be required to dissect this temporal regulation and fully elucidate the interplay between Sgo1-PP2A, ATRX, and Wapl in the protection of centromeric cohesion”.

It should also be noted that although ATRX plays a less prominent role than Sgo1 in safeguarding sister chromatid cohesion under physiological conditions, it is possible that ATRX mutations disrupting its interaction with Pds5B, which may only mildly activate the spindle assembly checkpoint and thus allow cell survival, could contribute to chromosomal instability in tumors. In contrast, Sgo1 mutations that impair cohesin interaction are more likely to strongly activate the spindle assembly checkpoint and induce mitotic catastrophe and cell death. We have therefore included the following statement in the Discussion: “Future studies should investigate whether mutations in ATRX disrupt its interaction with Pds5B and how such changes might contribute to chromosomal instability in cancer.”

Prof. Fangwei Wang
Zhejiang University
Life Sciences Institute
866 Yuhangtang Rd
Nano Building Rm 577
Hangzhou, Zhejiang 310058
China

13th May 2025

Re: EMBOJ-2025-120195R
A chromatin-remodeling-independent role for ATRX in protecting centromeric cohesion

Dear Fangwei,

Thank you for submitting your final revised manuscript for our consideration. I am pleased to inform you that we have now accepted it for publication in The EMBO Journal.

With kind regards,

Hartmut

Referee #1:

Although I was quite pleased with the previous version of this paper, I had still raised a number of issues. In their revised manuscript, Dr Wang and co-workers have adequately addressed all (but one) of the fourteen points of criticism, many of them by performing additional experiments. The new data fully support and further strengthen the main findings. The only minor point of criticism that remains is that the work of Hellmuth and Stemmann (2024) is misquoted when the authors write that "Wapl becomes globally inactive during metaphase". Apart from this, this is now an impressive paper that is truly worthy of publication. [Editor's note: I have deleted the reference solely on this occurrence]

Referee #2:

I have reviewed the authors' point-by-point response and find that they have satisfactorily addressed my previous concerns. I have no further comments and recommend that the paper be accepted for publication.
